# Two-dimensional electronic transport and surface electron accumulation in MoS$_2$

M.D. Siao[1], W.C. Shen[2], R.S. Chen [1], Z.W. Chang[3], M.C. Shih[4], Y.P. Chiu [3,4] & C.-M. Cheng[5,6]

Because the surface-to-volume ratio of quasi-two-dimensional materials is extremely high, understanding their surface characteristics is crucial for practically controlling their intrinsic properties and fabricating p-type and n-type layered semiconductors. Van der Waals crystals are expected to have an inert surface because of the absence of dangling bonds. However, here we show that the surface of high-quality synthesized molybdenum disulfide (MoS$_2$) is a major n-doping source. The surface electron concentration of MoS$_2$ is nearly four orders of magnitude higher than that of its inner bulk. Substantial thickness-dependent conductivity in MoS$_2$ nanoflakes was observed. The transfer length method suggested the current transport in MoS$_2$ following a two-dimensional behavior rather than the conventional three-dimensional mode. Scanning tunneling microscopy and angle-resolved photoemission spectroscopy measurements confirmed the presence of surface electron accumulation in this layered material. Notably, the in situ-cleaved surface exhibited a nearly intrinsic state without electron accumulation.

[1] Graduate Institute of Applied Science and Technology, National Taiwan University of Science and Technology, Taipei 10607, Taiwan. [2] Department of Electronic Engineering, National Taiwan University of Science and Technology, Taipei 10607, Taiwan. [3] Department of Physics, National Taiwan Normal University, Taipei 11677, Taiwan. [4] Department of Physics, National Taiwan University, Taipei 10617, Taiwan. [5] National Synchrotron Radiation Research Center, Hsinchu 30076, Taiwan. [6] Department of Physics, National Sun Yat-Sen University, Kaohsiung 80424, Taiwan. Correspondence and requests for materials should be addressed to R.S.C. (email: rsc@mail.ntust.edu.tw)

The metallic nature of graphene[1,2] differs from that of two-dimensional (2D) nanostructures based on transition metal dichalcogenide (TMD) layer materials, such as $MoS_2$, $WS_2$, and $ReS_2$. In addition, these 2D nanostructures exhibit semi-conducting characteristics and have thus attracted considerable attention[3–6]. The discovery of an electronic structure transition from an indirect to a direct bandgap in $MoS_2$ monolayers has opened up a new direction for photonic and optoelectronic applications involving TMD layer materials[7,8]. Quasi-2D semi-conductors are ideal systems to integrate with graphene[9] for the development of next-generation, ultrathin, flexible, transparent, light emitting[10], light-harvesting[11,12], and light-detecting devices[13]. A remarkably high on/off ratio of up to $10^8$–$10^9$ was also realized in 2D $MoS_2$ field-effect transistors (FETs)[14–17]. In addition, other applications, such as photocatalysts[18,19], batteries[20,21], solar cells[22], transistors[23], and memory devices[24,25], have been developed using 2D TMD-graphene hybrids as building blocks.

With an increasing number of studies concentrating on different applications of TMD nanomaterials, understanding the surface effect on electronic transport properties has become crucial. The 2D electron concentration ($n_{2D}$) in monolayer[15,24] and multilayer[26,27] $MoS_2$ FETs can easily reach $2 \times 10^{13}$–$1 \times 10^{15}$ $cm^{-2}$ under a gate voltage. The high on/off ratio and saturation current of 2D $MoS_2$ FETs operated in depletion mode partially benefit from the high electron concentration. However, the residual concentration at zero gate voltage ($V_g$) of the $MoS_2$ monolayers[15] at $5.6 \times 10^{12}$ $cm^{-2}$ is much higher than that ($\sim 1.6 \times 10^{10}$ $cm^{-2}$) of their bulk counterparts[28]. Novoselov et al. also observed $MoS_2$ monolayers with an anomalously high doping level ($n_{2D} = 10^{12}$–$10^{13}$ $cm^{-2}$) as the semi-metallic $NbSe_2$ monolayers[29].

The unusually high electron concentration in the unintentional doped layer semiconductor increases the difficulty of fabricating intrinsic and p-type $MoS_2$ nanostructures. Although p–n junctions have been used in heterostructures such as $p$-$WSe_2$/$n$-$MoS_2$[30] and $p$-$GaTe$/$n$-$MoS_2$[31], $MoS_2$ can only be used as an n-type component. Understanding the origin of heavy n-doping is critical for practical control of the conducting type and carrier concentration of $MoS_2$ 2D structures.

Van der Waals crystals without dangling bonds, such as $MoS_2$, are expected to have an inert surface and fewer surface states. Nevertheless, researchers have reported that electric contacts between Schottky metals and $MoS_2$ differ considerably from their expected characteristics[14,16,32–38]. An anomalously low Schottky barrier height is probably attributable to Fermi-level pinning and interface states[32,38].

In this study, we demonstrated that the pristine surface exhibits a remarkably high electron concentration and can be the origin of anomalously high n-doping in $MoS_2$ nanostructures. This surface characteristic results in surface-dominant and thickness-dependent electronic transport in the $MoS_2$ flakes. We proposed a transfer length method (TLM) based on 2D transport to support the preceding statement. Scanning tunneling microscopy/ spectroscopy (STM/STS) and angle-resolved photoemission spectroscopy (ARPES) characterizations provide direct evidence of the presence of surface electron accumulation (SEA) in $MoS_2$ single crystals. In addition, we observed that the in situ-cleaved fresh surface exhibits a nearly perfect intrinsic property without electron accumulation. The electrons accumulate gradually at the surface due to desulfurization at room temperature and even at low temperatures. This understanding enables us to achieve quasi-intrinsic $MoS_2$ devices by surface protection. The FET using quasi-fresh $MoS_2$ nanoflakes exhibit a much higher mobility and lower concentration of electrons than those with pristine surface. To date, limited material systems, such as $InAs$[39–42], $InN$[43,44], $CdO$[45,46], and $In_2O_3$[47], have been found to possess the SEA

characteristic. This study demonstrated the presence of SEA in TMDs and layered material systems.

## Results

**Characterization of CVT-grown $MoS_2$ crystals and FIB-fabricated nanoflake devices.** Figure 1a depicts the scanning electron microscope (FESEM) images of $MoS_2$ layer crystals on a dicing tape after preliminary mechanical exfoliation. The micrographs show that the stripped flakes have a random shape and that some of them retained clear sharp edges. The area size of a stripped flake (approximately less than $5 \times 5$ $\mu m^2$) is quite smaller than that of bulk crystals (several $mm^2$). A chemical vapor transport (CVT)-grown bulk layer crystal of $MoS_2$ is presented in the insets of Fig. 1a. Figure 1b illustrates the X-ray diffractometry (XRD) patterns for the vertically aligned $MoS_2$ bulk. As shown in the figure, four diffraction peaks centered at 14.3°, 29.0°, 44.1°, and 60.1° were detected, indexed as (002), (004), (006), and (008), respectively, of the $c$-planes of $MoS_2$ (JCPDS #872416). The single out-plane orientation shows the single-crystalline structure of the two-hexagonal (2H) $MoS_2$. Figure 1c depicts the Raman scattering measurement for $MoS_2$ bulk crystals. Two major Raman modes at 383.1 and 408.2 $cm^{-1}$ were consistent, respectively, with the $E^1_{2g}$ and $A_{1g}$ modes for 2H $MoS_2$[48]. The Raman peaks with very narrow peak widths at 3.1 ($E^1_{2g}$) and 3.7 ($A_{1g}$) $cm^{-1}$ further confirm the excellent crystalline quality of layer crystals.

The electric properties and ohmic contact of $MoS_2$ nanoflakes were examined using two-terminal $I$–$V$ measurements. Figure 1d depicts the $I$–$V$ curves for $MoS_2$ nanoflakes with different thicknesses. A common linear relationship of the $I$–$V$ curves indicates favorable ohmic contact of FIB-fabricated nanoflake devices. A chip template with a patterned Ti/Au multiple electrodes used for the $MoS_2$ nanoflake device fabrication by FIB is shown in the inset of Fig. 1d. The thickness values of $MoS_2$ flakes on chip templates were defined using atomic force microscopy (AFM) measurements. Figure 1e depicts a typical AFM image and the height profile of an $MoS_2$ nanoflake with a thickness of 22 nm.

**Thickness-dependent conductivity.** The electric conductance values of $MoS_2$ nanoflakes with different thicknesses were estimated using the slopes of the $I$–$V$ curves. Figure 2a shows that when the thickness increases for over one order of magnitude, the conductance did not substantially increase but remained constant. This observation contradicts the theoretical expectation. In principle, conductance ($G$) is linearly proportional to thickness, and conductivity ($\sigma$) is a dimension-independent constant for a uniform conductor and is written as:

$$G = \frac{I}{V} = \sigma \frac{A}{l} = \sigma \frac{wt}{l}, \qquad (1)$$

where $A$ is the area for current transport, and $l$, $w$, and $t$ are the length, width, and thickness of the conducting channel, respectively. For 2D-like crystals, $l$ and $w$ values are at the same micrometer scale and are several orders of magnitude higher than $t$ values. The near-constant value of $G$ implies a remarkable dependence of $\sigma$ on the thickness. To identify this point, $\sigma$ values were calculated using the Eq. (1).

Figure 2b depicts the log–log plot of $\sigma$ versus $t$ for $MoS_2$ flakes. The result reveals that the $\sigma$ value increased from 11 to 360 $\Omega^{-1}$ $cm^{-1}$, with a decrease in $t$ from 385 to 33 nm. The overall $\sigma$ values of nanoflakes were much higher than those ($\sigma \leq 0.1$ $\Omega^{-1}$ $cm^{-1}$) of their bulk crystals ($t \geq 10$ $\mu m$). The $\sigma$ range for bulks is defined using our measurements and reference values[28]. In addition, the $\sigma$ value is nearly inversely proportional to $t$. An

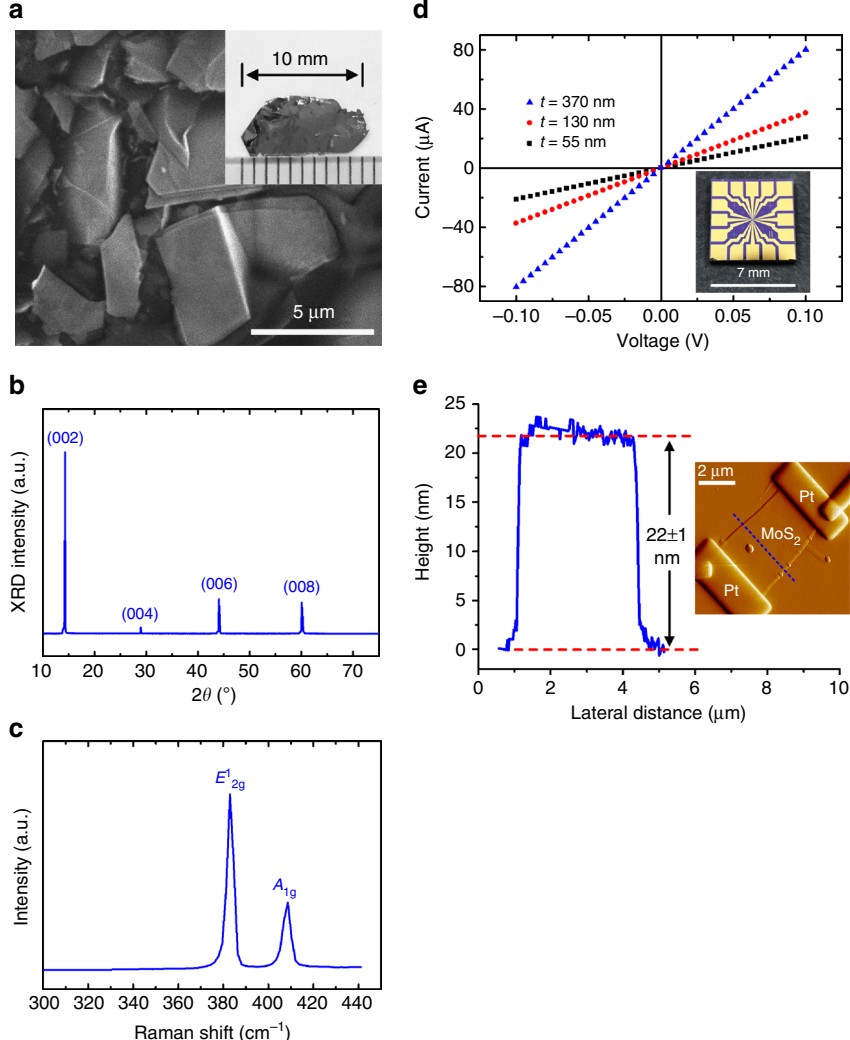

**Fig. 1** Characterization of CVT-grown $MoS_2$ crystals and FIB-fabricated nanoflake devices. **a** SEM image for $MoS_2$ flakes after preliminary mechanical exfoliation by using dicing tape. The inset in **a** depicts a photo of pristine bulk crystals of $MoS_2$. **b** XRD pattern and **c** Raman spectrum for $MoS_2$ bulk crystals grown using the CVT method. **d** I–V curves measured using the two-probe method for $MoS_2$ nanoflakes with different thicknesses. The inset in **d** depicts a chip template with patterned multiple Ti/Au electrodes used for the $MoS_2$ nanoflake device fabrication by FIB. **e** The height profile and image of an AFM measurement of a $MoS_2$ nanoflake with a thickness of 22 nm

inverse power law of $\sigma \propto t^{-\beta}$ could be obtained, where the fitted $\beta$ value is $1.1 \pm 0.16$.

**Temperature-dependent conductivity**. Temperature ($T$)-dependent conductivity measurements for $MoS_2$ nanoflakes ($t = 52$ nm) and bulk crystals ($t = 86$ μm) were also performed (Fig. 3a). To clearly present the difference between nanoflakes and bulk crystals, the $\sigma$ versus $T$ curves were normalized by their $\sigma$ values at 300 K. The result indicated that $MoS_2$ nanoflakes exhibit weak semiconducting behavior, which is different from that of their bulk counterparts. The $\sigma$ value of bulk crystals is considerably more sensitive to a temperature decrease from 300 to 180 K. Furthermore, the thermal activation energy of the majority carrier ($E_a$) could be obtained using the Arrhenius plot. The Arrhenius equation can be represented as follows: $\sigma(T) = \sigma_0 \exp(-E_a/kT)$, where $\sigma_0$ is the conductivity at $T = 0$ K and $k$ is Boltzmann's constant. The $E_a$ value can be obtained by fitting the Arrhenius slopes ($-E_a/1000k$) of the $\ln\sigma$ versus $1000/T$ curve. Figure 3b depicts the Arrhenius plots of the $\sigma-T$ curves for $MoS_2$ nanoflakes and bulk crystals. The results indicate that $E_a$ values of

nanoflakes are much smaller (6 meV) than those of bulk crystals (68 meV).

The different $E_a$ values denote the different origins of the majority carriers. Because nanoflakes were exfoliated from the same piece of a bulk crystal, their structural quality and chemical composition are theoretically the same. The nanoflakes and bulks with different doping concentrations and defect types can be eliminated. Accordingly, we inferred that the majority carriers in nanoflakes originated from the surface rather than the inner bulk. The presence of very shallow donor-like surface states or resonance surface states[40,43,49], which are energetically higher than the donor level in the bulk, could dominate the current transport in thin flakes.

**2D electronic transport defined by TLM**. To examine the probable surface-controlled conduction property and the contact resistance, a TLM model was adopted to analyze $MoS_2$ flakes with different thicknesses. According to the conventional 3D TLM model, total resistance ($R$) is equal to the sum of two contact resistance ($R_c$) and sample resistance ($R_s$) and is written as

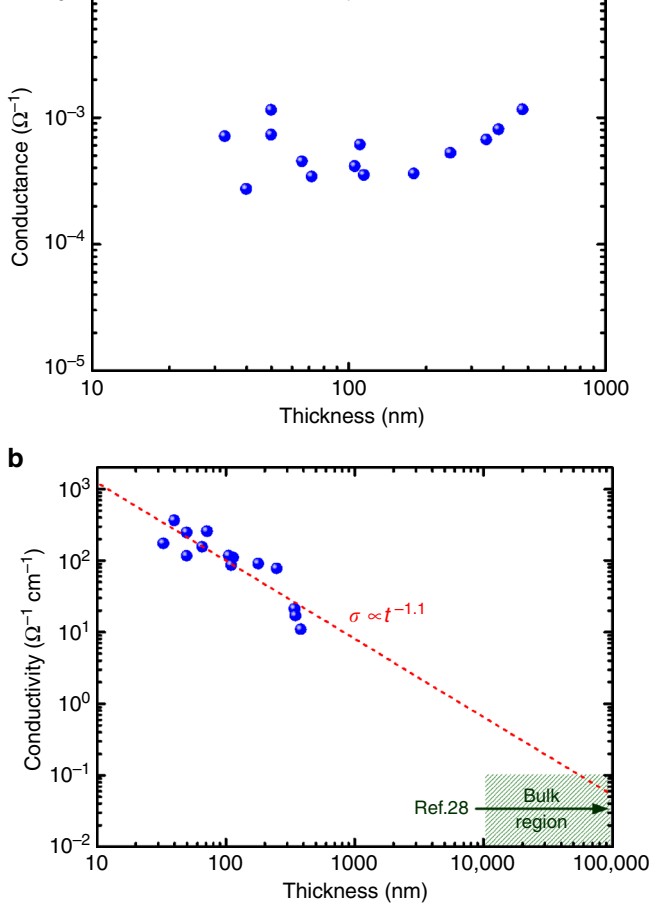

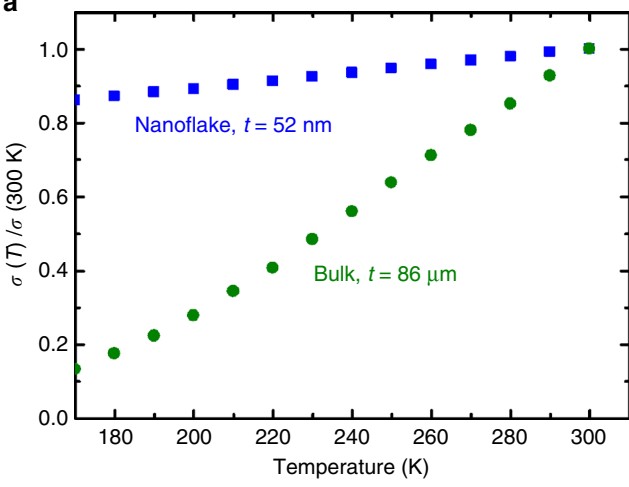

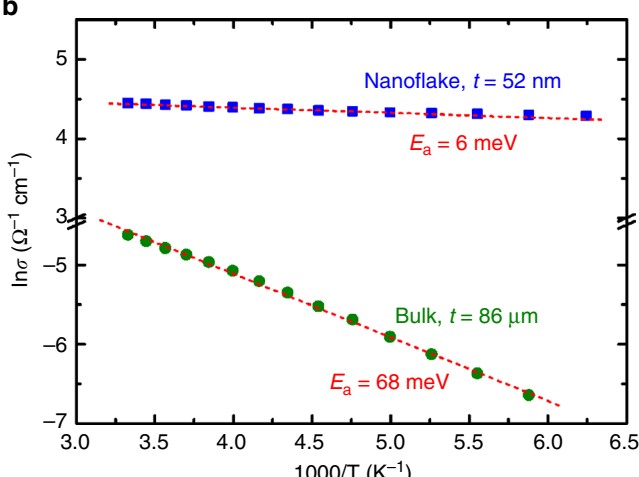

**Fig. 2** Thickness-dependent conductivity in MoS$_2$ nanoflakes. **a** Linear plot of conductance versus thickness and **b** logarithmic plot of conductivity versus thickness for MoS$_2$ nanoflakes. The conductivity range of MoS$_2$ bulks obtained by our measurements and from ref. [28] is shown for comparison. The red dashed line is the fitting line for the conductivity versus thickness data

**Fig. 3** Temperature-dependent conductivities in MoS$_2$ nanoflakes and bulks. **a** Temperature-dependent conductivity, $\sigma(T)$, curves and **b** their corresponding Arrhenius plots for an MoS$_2$ nanoflake ($t = 52$ nm) and a bulk crystal ($t = 86$ μm). The $\sigma(T)$ values of the MoS$_2$ nanoflake and bulk in **a** were normalized by their corresponding conductivity values at 300 K, $\sigma$(300 K), to present the difference in $\sigma$ between the nanoflake and the bulk. The red dashed lines show the fitting lines of the Arrhenius slopes for activation energy ($E_a$) calculation

follows:[50]

$$R = 2R_c + R_s = 2R_c + \rho_{3D}\frac{l}{wt}, \qquad (2)$$

where $\rho_{3D}$ is the resistivity of the 3D conductor. Theoretically, $R$ is linearly dependent on the dimension-related variable of $\frac{l}{wt}$. $R_c$ and $\rho_{3D}$ values can be obtained by fitting the intercept and slope of the $R$ versus $\frac{l}{wt}$ curve, respectively. However, the $R$ versus $\frac{l}{wt}$ plot shown in Fig. 4a does not exhibit a clear linear relationship. Besides the two data points at $\frac{l}{wt}$ higher than $7 \times 10^5$ cm$^{-1}$, most of the data points did not follow a linear trend. If intentionally conducting a linear fitting, an unreasonably low $\rho_{3D}$ at 0.002 Ω cm can be obtained, which is far from the general range of 10–100 Ω cm of MoS$_2$ bulks.

Considering the probable surface-controlled conduction in MoS$_2$, the current predominantly flows through the surface rather than the bulk. The near 2D current transport behavior could not follow the TLM model based on a uniform 3D conductor. Therefore, a modified TLM model based on 2D transport is proposed. Assuming an ideal 2D conductor without thickness, its resistance $R_s$ only depends on the length and width for current flow and is independent of thickness. The 2D TLM model can be

expressed as follows:

$$R = 2R_c + R_s = 2R_c + \rho_{2D}\frac{l}{w}, \qquad (3)$$

where $\rho_{2D}$ is the resistivity of the 2D conductor (or sheet resistance[51]). According to Eq. (3), the $R$ value is proportional to $l/w$. Figure 4b depicts the plot of $R$ versus $l/w$ curve for MoS$_2$ flakes. The overall $R$ values followed a linear relationship with $l/w$. The $R_c$ at 440 Ω and $\rho_{2D}$ at 450 Ω could also be obtained by fitting the intercept and slope of the $R$ versus $l/w$ curve, respectively. The TLM analysis clearly indicates that MoS$_2$ nanostructures follow a 2D transport behavior rather than a 3D mode. The result supports the aforementioned hypothesis of surface-dominant conduction in MoS$_2$.

Although MoS$_2$ nanoflakes are thicker than monolayers, the quasi-2D transport in flakes may allow a fair comparison of the specific contact resistance ($R_{sc} = R_c \times w$) among these 2D systems[52]. The typical $w$ values of our flake devices are 1–3 μm.

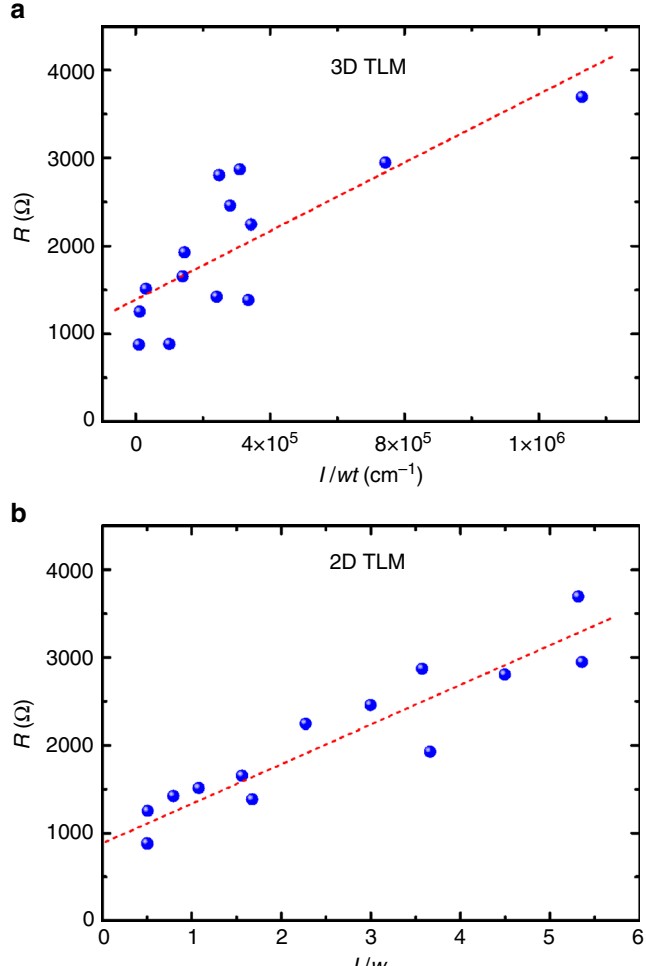

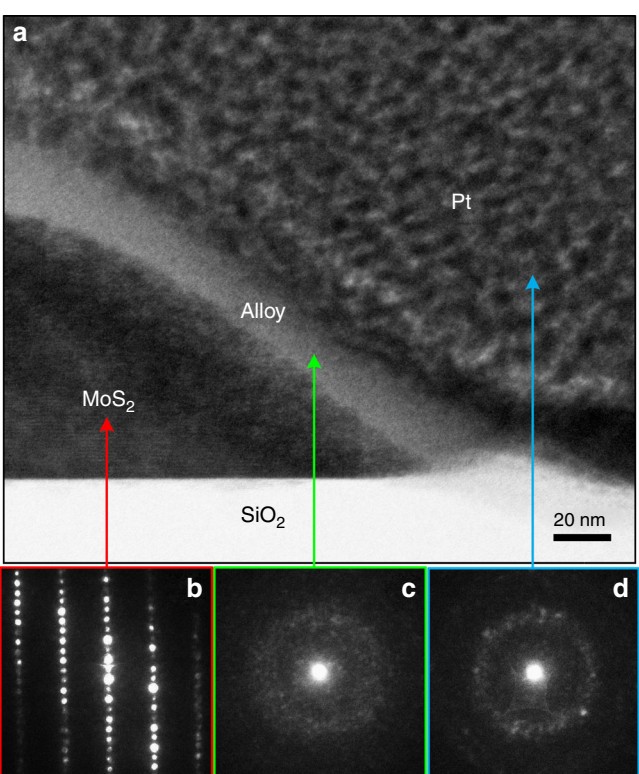

**Fig. 5** TEM characterization of Pt/MoS$_2$ interface. **a** TEM image focusing on the metal contact of the Pt/MoS$_2$ interface in an FIB-fabricated MoS$_2$ nanoflake device. The corresponding SAED patterns taken from the regions of **b** MoS$_2$, **c** alloy layer, and **d** Pt metal are also illustrated

**Fig. 4** Analysis of the transfer length method (TLM) for MoS$_2$ nanoflake devices. **a** 3D TLM plot and **b** 2D TLM plot for MoS$_2$ nanoflakes. The red dashed lines are the fitting lines for the resistance ($R$) versus $l/wt$ curve in **a** and $R$ versus $l/w$ curve in **b**

The corresponding $R_{sc}$ values are at 440–1320 Ω μm. These values are much lower than those of most metal contacts on MoS$_2$ such as Ti on monolayer, few-layer, and multilayer MoS$_2$ ($R_{sc}$ = 800 −740,000 Ω μm);[26,53,54] Ni/Au on few-layer MoS$_2$ ($R_{sc}$ = 18,000 Ω μm, $V_g$ = 0 V);[16] and Mo on few-layer MoS$_2$ ($R_{sc}$ ~2000 Ω μm, $V_g$ = 0 V)[52]. The lower bound of $R_{sc}$ at 440 Ω μm is also comparable with the optimal $R_{sc}$ values achieved using different contact approaches such as lateral metallic 1T MoS$_2$ contacted with 2H MoS$_2$ nanosheets ($R_{sc}$ = 200−300 Ω μm)[55], Ni/etched bilayer graphene on MoS$_2$ flakes ($R_{sc}$ = 460 Ω μm, $V_g$ = 0 V)[51], and Ni/Au on Cl-doped few-layer MoS$_2$ ($R_{sc}$ = 500 Ω μm)[56]. However, the FIB approach used to fabricate the Pt contact on MoS$_2$ is relatively simple because pretreatment of the contact surface, post annealing, and multilayer electrode fabrication are not necessary. The metal deposition area is selective, and ohmic contact is formed simultaneously with metal deposition through FIB processing. The formation of a conducting amorphous alloy layer in the Pt and MoS$_2$ interface could be the reason for the low contact resistance obtained using the FIB approach, as is discussed in the following section.

**Metal/semiconductor electric contacts characterized by TEM.**
Layer semiconductors, such as MoS$_2$[57], MoSe$_2$[58], and ReS$_2$[59],

exhibit strong anisotropic transport properties. The in-plane conductivity perpendicular to the $c$-axis (i.e., $\sigma_\perp$) is over three orders of magnitude higher than that along the $c$-axis (i.e., $\sigma_{//}$). Because in-plane transport is much easier than interlayer transport, electric current preferentially flows along the channel close to the upper surface if metal electrodes are only constructed on the sample surface. The anisotropic electric property can produce results similar to those of surface-dominant transport. A simulation study indicated that the current flow can be limited to a few specific layers in an FET device with a top-contact configuration[60].

To rule out this possibility and ensure a uniform current flow condition at a defined cross-sectional area, metal contacts including the top surface and side walls in particular were entirely covered by Pt (Fig. 5). The cross-sectional transmission electron microscope (TEM) image shows that deposited Pt completely covered the side wall and partial top surface of the MoS$_2$ nanoflake with a thickness of approximately 100 nm. An alloy layer with a thickness of 20–30 nm formed by ion bombardment during Pt deposition was also observed at the interface of Pt/MoS$_2$. The alloy mixed with Pt, Mo, and S (confirmed using energy-dispersive X-ray spectroscopy (EDX) analysis; not shown here) with an amorphous structure was confirmed using selected-area electron diffractometry (SAED) measurements (Fig. 5c). The amorphous buffer layer can minimize the effects of surface contaminants and the Schottky barrier on the Pt/MoS$_2$ interface and thus facilitate the formation of ohmic contact. The well-contacted side walls of flakes can ensure uniform current flow condition and prevent the anisotropic effect on 2D transport. Similar results have also been observed for FIB-fabricated MoSe$_2$ devices[61,62].

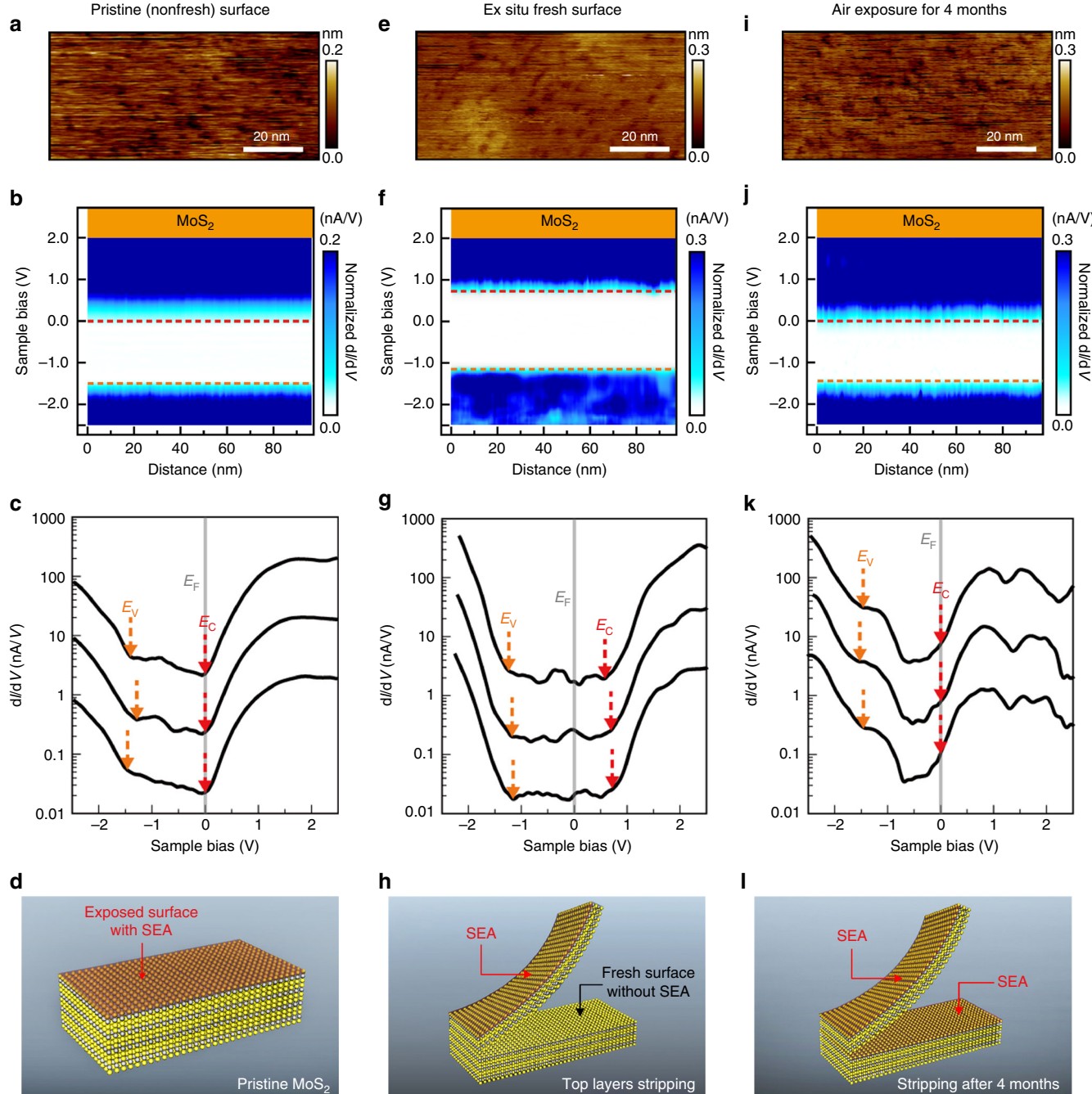

**Fig. 6** STM/STS characterization and surface electron accumulation (SEA) of a MoS$_2$ single crystal. **a** STM image, **b** band alignment, and **c** d$I$/d$V$ curve measurements for the non-fresh surface of a pristine MoS$_2$ crystal. **d** Schematic SEA of pristine MoS$_2$. **e** STM image, **f** band alignment, and **g** d$I$/d$V$ curve measurements for the fresh surface of an MoS$_2$ crystal. **h** Schematic shows the fresh surface of MoS$_2$ without SEA and the original non-fresh surface with SEA. **i** STM image, **j** band alignment, and **k** d$I$/d$V$ curve measurements for the fresh surface of an MoS$_2$ crystal exposed to air for four months. **l** Schematic showing the SEA formation at the fresh surface of MoS$_2$ after four months exposure to air

**Surface electron accumulation in MoS$_2$ evidenced by STM and STS.** To obtain direct evidence of the high surface electron concentration, surface probing by using STM was performed. Figure 6 illustrates the STM measurements of "pristine" (original exposure surface before stripping, Fig. 6a–d) and "fresh" surfaces (produced by top-layer stripping by using dicing tape in Fig. 6e–h) in MoS$_2$ single crystals. Figure 6c shows d$I$/d$V$ versus the sample bias ($V$) curves obtained at a pristine (non-fresh) surface. The onsets of the normalized d$I$/d$V$ curves at positive and negative sample biases corresponded to the conduction band edge

($E_c$) and valence band edge ($E_v$), respectively. The Fermi-level position (at $V = 0$ V) was located very near to the conduction band edge for the pristine surface. This result confirms that the MoS$_2$ surface is highly n-doped. Moreover, the band alignment diagram (Fig. 6b) shows that the energy difference between the Fermi level and conduction band minimum is relatively uniform over the whole probing range (~100 nm), indicating that the metal-like surface is not confined to a few local defect areas. We probed several positions and noticed that the n-doping region is typical of the whole sample (area >5 × 5 mm$^2$). Unlike in natural

$MoS_2$ crystals, which are relatively defective, as reported by McDonnell et al.[38], a p-doping region was not observed in our CVT-grown $MoS_2$.

According to the energy difference ($\Delta E$) between the conduction band minimum ($E_c$) and Fermi level ($E_F$), the electron concentration ($n$) in the $MoS_2$ surface can be estimated using the following equation:

$$n = N_c \exp\left[\frac{-(E_c - E_F)}{kT}\right], \qquad (4)$$

where $N_c = 2(2\pi m_e^* kT/h^2)^{3/2}$ is the effective density of states in the conduction band, $m_e^*$ is the effective electron mass[63], $k$ is Boltzmann's constant, $T$ is the temperature set at 300 K, and $h$ is Planck's constant. Because the Fermi level nearly overlaps the conduction band edge, $\Delta E$ is approximately zero (i.e., $n \sim N_c$). According to the literature, $m_e^*$ is in the range of 0.45–0.73 $m_0$ for $MoS_2$ bulk materials, where $m_0$ is the free electron rest mass[64]. The calculated electron concentration on the "pristine" surface ($n_s$) is in the range of $7.5 \times 10^{18}$–$1.6 \times 10^{19}$ cm$^{-3}$, which is nearly four orders of magnitude higher than the value ($n = 2 \times 10^{15}$ cm$^{-3}$) of the bulk crystal with a thickness of 2 mm[28]. The four orders of magnitude difference in the electron concentration between the surface and bulk are also consistent with the conductivity difference observed in the thickness-dependent conductivity (Fig. 2b).

Because the pristine (non-fresh) surface (Fig. 6a–c) was exposed to air for a prolonged period following crystal growth, the fresh surface that was obtained by stripping the top layers using dicing tape was also measured. Figure 6e–g depicts the STM measurement of the fresh surface. Notably, the fresh surface did not exhibit metal-like characteristics. According to the d$I$/d$V$ versus $V$ curves, the $E_F$ position is located far from the $E_c$ (Fig. 6g). This result indicates that the fresh surface has a considerably lower electron concentration and is likely intrinsic, differing from the metallic non-fresh surface that has been exposed to air. Estimating the electron concentration of the fresh surface is difficult because the $\Delta E$ value could be highly overestimated owing to the tip-induced band bending (TIBB) effect of the relatively insulating surface. In addition, the band alignment diagram also shows that the intrinsic area is uniform in the probing region (~100 nm) (Fig. 6f). The remarkable difference between the metallic non-fresh surface and the insulating fresh surface observed using STM is also consistent with the several orders of magnitude difference in the electron concentration between the surface (estimated by our STM measurement) and bulk crystals (according to ref. [28]) in the aforementioned discussion because the fresh surface originates from the inner layers of the bulk.

The bandgap determined by our STS measurement for the non-fresh $MoS_2$ is approximately 1.35 ± 0.05 eV. Due to the TIBB effect in the insulating fresh surface, the bandgap is overestimated at 1.8−1.9 eV. Lu et al. also investigated the bandgap of bulk $MoS_2$ by STS, and their statistic bandgap values are at 1.29 ± 0.045 eV[65]. The lower bound of our measured bandgap at 1.28 eV is consistent with the mean value in the ref. [65].

The term "fresh surface" is nominal because the new surface after top-layer stripping was exposed to the air for less than 30 min during preparation before being loaded into the STM chamber. Nevertheless, the fresh surface still maintained an intrinsic-like state, implying that surface doping due to air exposure is not a rapid process. The same surface was measured again using STM after 4 months of exposure to air (Fig. 6i–k). Notably, the $E_F$ position shifted to near the conduction band (Fig. 6k) indicating a metallic character. The band alignment diagram in Fig. 6j exhibits uniform doping in the 100-nm probing range. This result indicates that the intrinsic fresh surface was transformed into a metallic surface by long-term exposure to air.

## Surface electron accumulation and low-temperature desulfurization evidenced by ARPES

To obtain a mechanistic understanding of the probable origin of the SEA, the electronic structure of the surface of the $MoS_2$ single crystals was further characterized by ARPES. Figure 7a–c illustrates the valance band for the pristine, ex situ, and in situ fresh surfaces of $MoS_2$ recorded with a 42-eV photon energy at 85 K. Here, the "ex situ" and "in situ" fresh surfaces are defined as the fresh surfaces cleaved in ambient air (exposure time less than 10 min) and in ultra-high vacuum (UHV) ($< 1.8 \times 10^{-10}$ Torr), respectively. The band mapping result shows a clear shift in the valance band maximum (VBM) relative to the Fermi level when the surface condition changes from pristine via the ex situ fresh to the in situ fresh state. The normal emission spectra at the $\Gamma$ point with different surface conditions are depicted in Fig. 7d, e. The spectrum shows a very sharp valance band edge at −0.75 eV binding energy in the in situ fresh surface. According to the bandgap at 85 K ($> 1.3$ eV), the corresponding Fermi level position is close to the middle of the bandgap (intrinsic Fermi level). Both the observations indicate that the new surface cleaved in the UHV has a nearly perfect structural quality and highly insulating nature. Once the fresh surface contacts the atmospheric ambience, the band edge exhibits a remarkable redshift to a binding energy of −0.98 eV for the ex situ fresh surface even though the exposure time is controlled to within 10 min. Long-term air exposure (pristine surface) can further decrease the band edge to −1.12 eV binding energy. The substantial valance band shift stands for the Fermi level with a blueshift for 0.37 eV. The higher Fermi-level position indicates a much higher electron concentration in the pristine surface. The result is consistent with the STM measurement, further confirming the presence of SEA in the $MoS_2$ crystals.

A band tail can also be observed accompanying the redshift of the valance band for the surfaces with air exposure (Fig. 7e). The presence of the band tail indicates the formation of structural defects or the incorporation of impurities in the $MoS_2$ surface. The exposure of the sulfide surface in air could result in two major effects, namely, the escape of sulfur atoms and the adsorption of foreign molecules. To clarify the origin of the donor-like surface states in $MoS_2$, the in situ-cleaved surface of $MoS_2$ was also measured under two different conditions by ARPES. Figure 8a, b depicts the valance band measurements for the in situ-cleaved surface and the surface kept at the same temperature of 85 K for 11 h. The corresponding normal emission spectra at the $\Gamma$ point (Fig. 8d, e) show a clear shift in the VBM from −0.80 to −1.11 eV. Surprisingly, the VBM or Fermi level exhibits such a significant shift for 0.31 eV simply due to the exposure at $5.6 \times 10^{-11}$ Torr base pressure with 11 h at 85 K. The result can exclude the possibility of SEA induced by the adsorption of foreign molecules, such as oxygen and water molecules.

Accordingly, sulfur vacancies due to the escape of sulfur atoms from the $MoS_2$ surface are likely the most prominent type of surface defect resulting in the SEA phenomenon. To further support this point, the in situ-cleaved surface was also annealed at 110 °C for 20 min. The valance band (Fig. 8c) and its corresponding normal emission spectrum (Fig. 8d, e) indicate that the VBM further shifts to the energy position at −1.22 eV. The result shows that the Fermi level can be easily pushed to the near conduction band edge by speeding up the desulfurization rate at a relatively high temperature. Furthermore, the gap states appearing at the near-valence band edge were also observed after annealing treatment, as shown in the inset of Fig. 8e. A similar band edge shoulder or band tail can also be observed for the pristine and ex situ-cleaved surfaces (Fig. 7e), which indicates that the SEA induced by long-term air exposure originates from the same sulfur vacancy surface defect.

A similar blueshift of the Fermi level due to desulfurization at UHV was also observed at room temperature by the ARPES

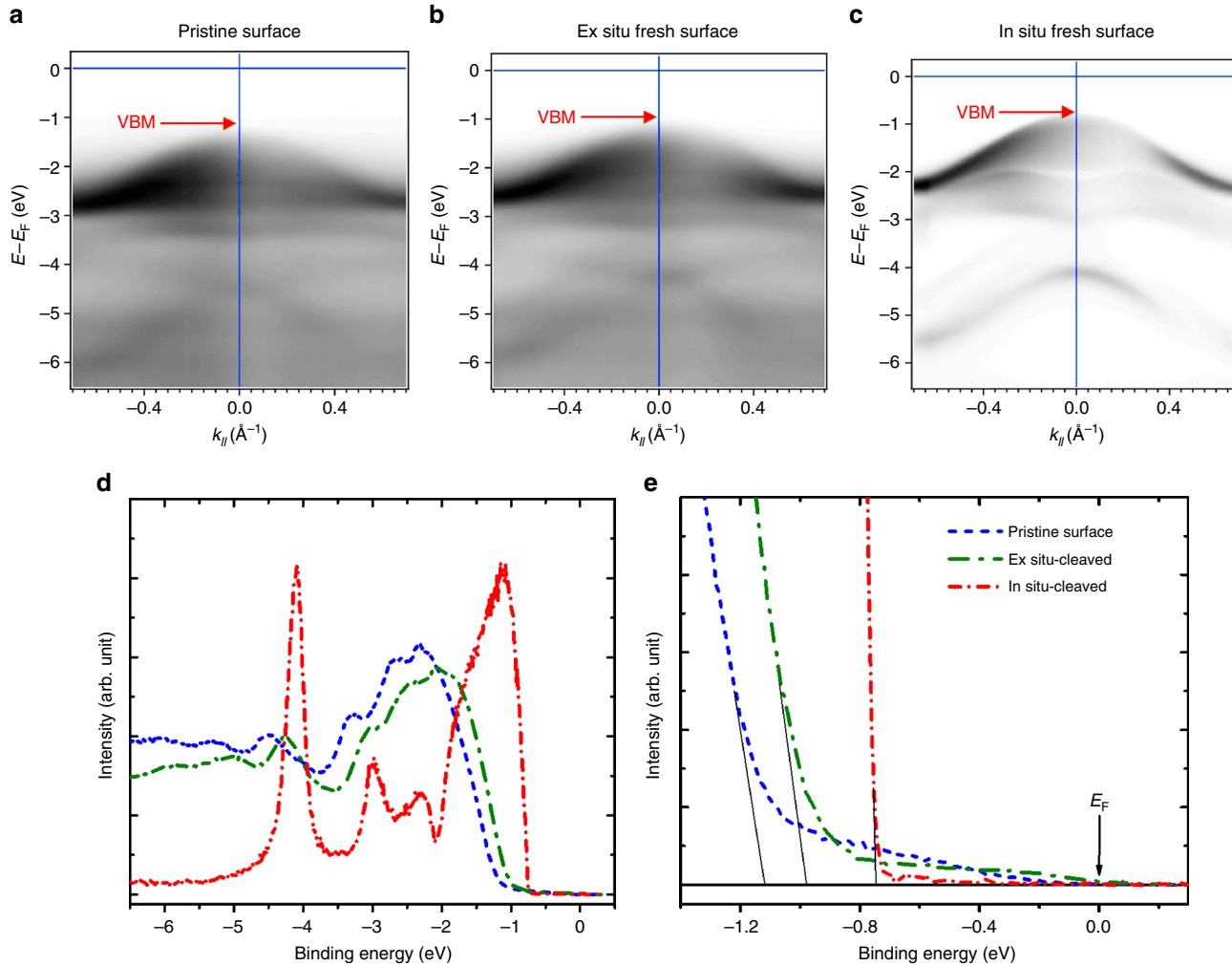

**Fig. 7** ARPES characterization of pristine, ex situ and in situ-cleaved fresh surfaces of a $MoS_2$ single crystal. The $E$ versus $k_{\parallel}$ valance band measurements for the **a** pristine, **b** ex situ, and **c** in situ-cleaved fresh surfaces of a $MoS_2$ crystal recorded with a 42-eV photon energy at 85 K. **d**, **e** The normal emission spectra at the $\Gamma$ point with different binding energy scale for the pristine, ex situ fresh and in situ fresh surfaces

measurement (see Supplementary Fig. 1). Thermogravimetric analysis (TGA) shows that $MoS_2$ is stable and that desulfurization is not significant at room temperature[66,67]. However, our result indicates that the van der Waals crystal surface without dangling bonds is not as stable as initially expected. Desulfurization occurs continuously at room temperature and even at low temperatures once a fresh surface has been created. Theoretical simulation predicted that the sulfur vacancy with the lowest formation energy among all the surface defect types can form spontaneously[68,69]. The previous STM and STS studies further indicated that sulfur deficiencies could be the origin of n-doping in $MoS_2$[38]. These results are consistent with our observations.

In addition, direct observation of sulfur vacancies in the $MoS_2$ surface has also been conducted by STM and STS. Sulfur vacancies introducing defect states in the bandgap close to the VBM have been reported previously[70–73]. An extra feature near the VBM measured at the defect center was observed in our STS spectra, which indicated the presence of sulfur vacancies in the $MoS_2$ surface. The results are detailed in Supplementary Figs. 2 and 3 and Supplementary Notes 1 and 2.

**Enhanced mobility in fresh $MoS_2$ nanostructures defined by FET measurement**. Figure 9 depicts the FET measurements for the $MoS_2$ nanoflakes with pristine and ex situ fresh surfaces. To minimize the influence of air exposure, a 50 nm $SiO_2$ layer was

coated on the surface of the "fresh" nanoflakes after exfoliation (Fig. 9c). The air exposure time was controlled to be less than 30 min for the ex situ fresh nanoflakes. Figure 9b, d depicts the drain to source current ($I_{ds}$) versus the gate voltage ($V_g$) curves for the pristine and fresh nanoflake FETs, respectively. The $I_{ds}-V_g$ curves show the n-type conduction characteristic for both types of $MoS_2$ flakes. Though there exist two electron accumulation layers at the top and bottom surfaces of the pristine $MoS_2$ flake, the variation of $I_{ds}$ induced by the gate voltage is dominated by the bottom surface (the $SiO_2/MoS_2$ interface). The field effect on the top surface (the air/$MoS_2$ interface), which is far away from the gate electrode, is negligible for the flake thicker than 50 nm, as in our case. Accordingly, the mobility can be calculated according to the conventional metal-oxide-semiconductor (MOS) FET model[14]

$$\mu = \left(\frac{dI_{ds}}{dV_{ds}}\right)\left(\frac{l}{wC_iV_{ds}}\right), \qquad (5)$$

where $l$ and $w$ are the length and width of the conducting channel, respectively, $C_i = 1.15 \times 10^{-8} \, \text{F cm}^{-2}$ is the capacitance between the channel and the back gate per unit area and can be calculated by $C_i = \varepsilon_0 \varepsilon_r/d$, where $\varepsilon_0 = 8.85 \times 10^{-14} \, \text{F cm}^{-1}$ is the permittivity of free space, $\varepsilon_r = 3.9$ is the dielectric constant of $SiO_2$, and $d = 300$ nm is the thickness of the bottom-gate oxide. $V_{ds} = 0.1$ V is the drain to source voltage. In addition, the linear $I_{ds}-V_{ds}$ curves exclude the

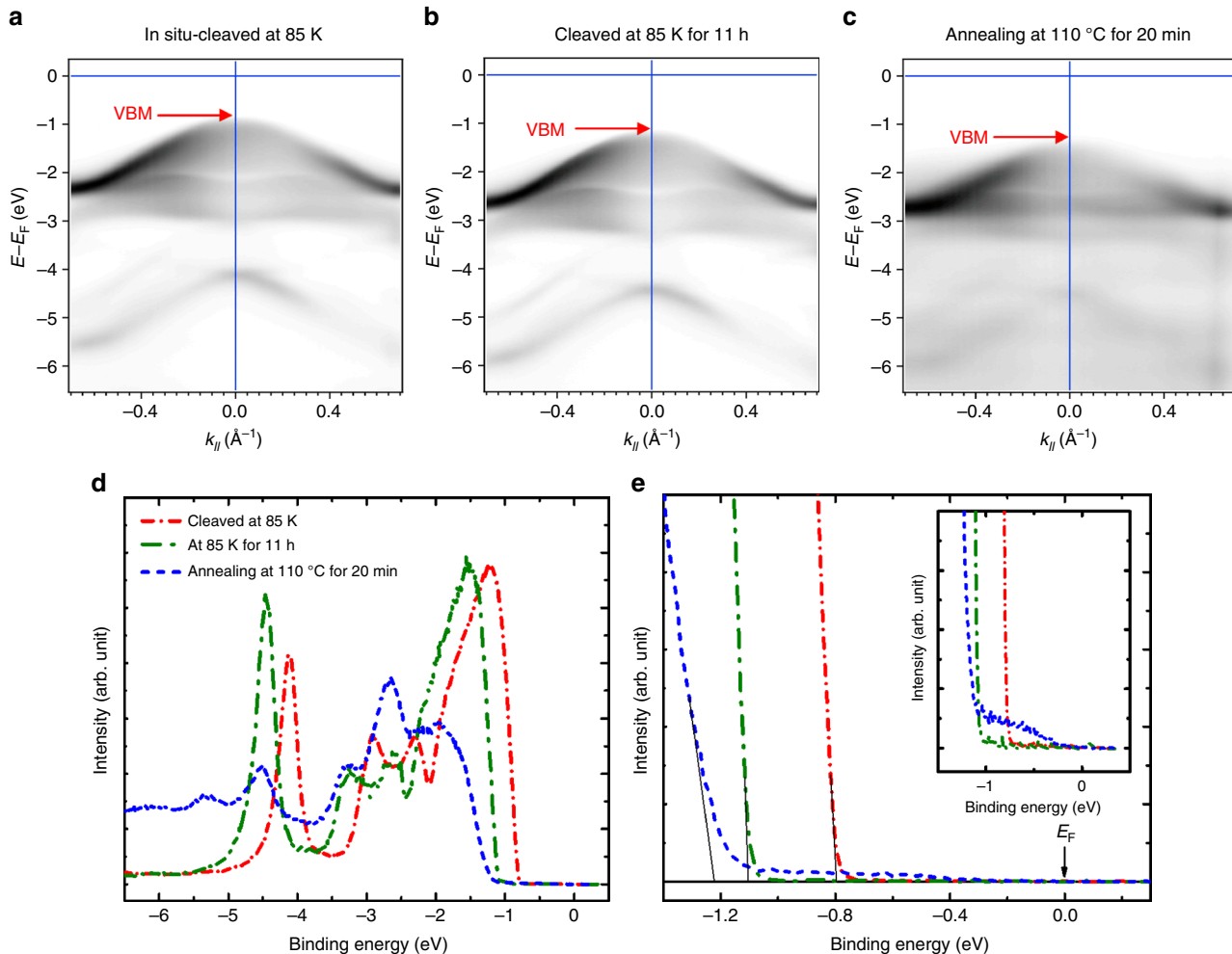

**Fig. 8** Aging effect at low temperature and annealing effect of the in situ-cleaved fresh surfaces of a MoS$_2$ single crystal. The $E$ versus $k_{\parallel}$ valance band measurements for the **a** in situ-cleaved surface at 85 K, **b** the in situ-cleaved surface at 85 K for 11 h, and **c** the in situ-cleaved surface annealed at 110 °C for 20 min of an MoS$_2$ crystal recorded with a 42-eV photon energy at 85 K. **d**, **e** The normal emission spectra at the Γ point with different binding energy scales for the in situ-cleaved surface at 85 K, the in situ-cleaved surface at 85 K for 11 h, and the in situ-cleaved surface annealed at 110 ºC for 20 min.

probable field effect induced by the Schottky contact at the source and drain electrodes (insets, Fig. 9b, d).

The result shows that the fresh nanoflakes exhibit a much higher mobility at 7.2 cm$^2$ V$^{-1}$ s$^{-1}$, which is nearly 30 times higher than that ($\mu = 0.25$ cm$^2$ V$^{-1}$ s$^{-1}$) of the pristine flakes. The enhanced mobility in the fresh flakes can be explained by the absence of surface scattering. Electronic transport in the pristine flakes has been confirmed to be dominated by the SEA. Because the surface is a defective area, the accumulated electron transport along the 2D channel inevitably suffers from substantial defect or impurity scattering. The surface scattering decreases the mobility of the pristine flakes. The mobility of the fresh flakes without surface-dominant scattering is comparable to the optimal values of the MoS$_2$ monolayers ($\mu = 0.1-10$ cm$^2$ V$^{-1}$ s$^{-1}$)[14], few layers and multilayers ($\mu = 2.1-26$ cm$^2$ V$^{-1}$ s$^{-1}$)[16,17,27] using the same bottom-gate FET configuration.

The comparison of the field-effect mobility between the fresh and non-fresh samples indicates substantial surface scattering in MoS$_2$, which is relatively consistent with the previous report[65]. Lu et al. reported the thickness-dependent mobilities for the MoS$_2$ nanoflakes (1–15 layers). The scattering from the surface defects was proposed to be a dominant factor determining the monotonic increase in mobility with the layer number.

The electron concentration of the fresh flakes is on the order of 10$^{17}$ cm$^{-3}$, which is much lower than that (>10$^{19}$ cm$^{-3}$) of the pristine ones. According to the ARPES result, the in situ fresh flake can possess an even lower electron concentration than the ex situ fresh sample used for the FET measurement. The result demonstrates that intrinsic MoS$_2$ nanostructures can be easily achieved by material processing in a vacuum or in inert gas surroundings, which is crucial for practical device applications and the realization of p-doping of MoS$_2$.

## Discussion

Figure 10 shows a schematic of the surface band diagram and electron accumulation that occurred due to the presence of donor-like surface states in MoS$_2$. The surface band bending induced by surface states is similar to that induced by the n–n$^+$ junction. The surface state density ($D_{ss}$) distribution was also plotted schematically to present Fermi-level pinning by surface states. The probable Fermi-level pinning could explain the common observation of the anomalously low Schottky barrier height for high work function metals in contact with the MoS$_2$[14,16,32–38]. The presence of the donor-like surface states may alter the ideal metal–semiconductor contact. Under these circumstances, the carriers that transfer from a semiconductor to a metal in a Schottky contact to equilibrate the Fermi level are mostly provided by the surface states rather than the semiconductor bulk. This mechanism reduces the loss of the carrier at the

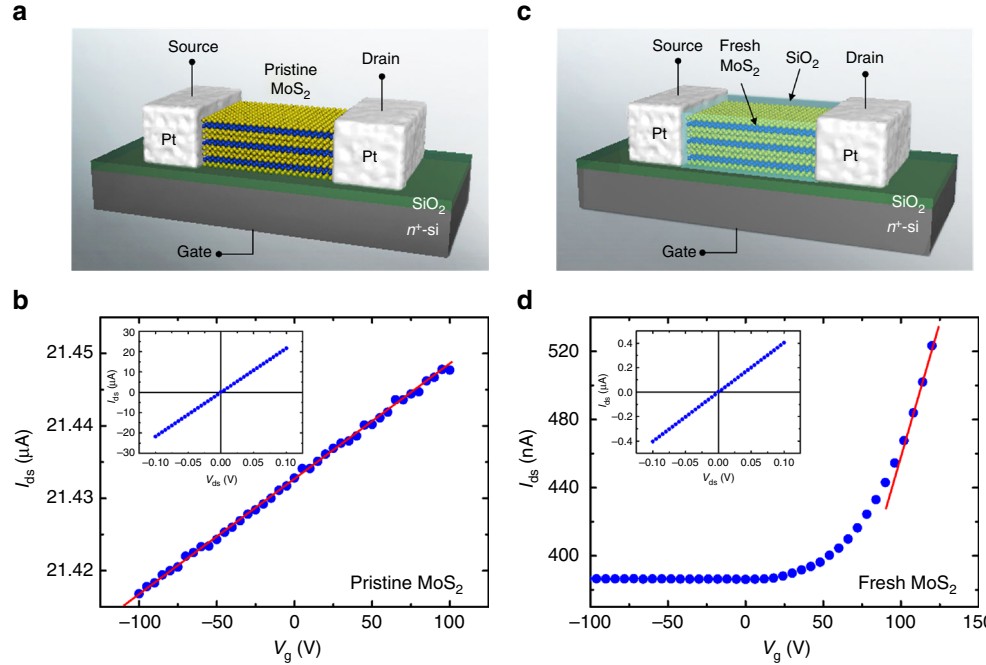

**Fig. 9** FET measurements of the pristine and fresh $MoS_2$ nanoflakes. The schematic FET device with a bottom-gate configuration and $I_{ds}$-$V_g$ curves for the **a–b** pristine and **c–d** fresh $MoS_2$ nanoflakes. The $V_{ds}$ is 0.1 V. The dimensions of the nanoflakes are $l = 4.3\,\mu m$, $w = 2.3\,\mu m$, and $t = 52\,nm$ for the pristine sample and $l = 5.3\,\mu m$, $w = 2.1\,\mu m$, and $t = 250\,nm$ for the fresh sample

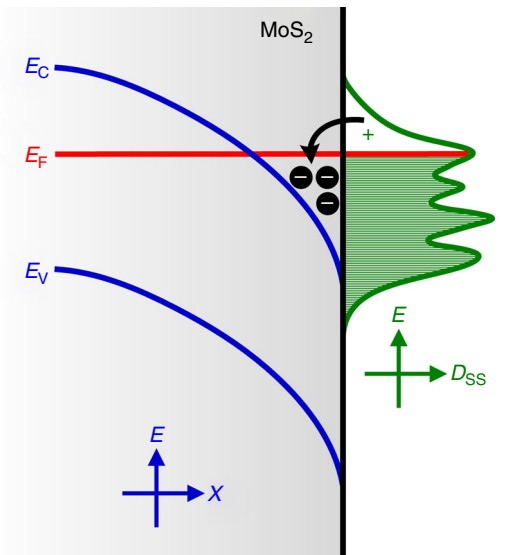

**Fig. 10** Schematic band bending and electron accumulation at a $MoS_2$ surface. A schematic band diagram of the surface of $MoS_2$. The surface band bending and surface electron accumulation (SEA) induced by the presence of donor-like surface states are illustrated. The surface state density ($D_{ss}$) distribution is plotted schematically to present the Fermi-level pinning by surface states

semiconductor site and the formation of the depletion region, thus lowering the Schottky barrier height.

In previous FET studies, monolayer and multilayer $MoS_2$ FETs were always operated in depletion mode[14–17,27,74,75]. An unusually high residual conductance or 2D electron concentration ($n_{2D}$) at $2 \times 10^{13}$–$6.5 \times 10^{15}\,cm^{-2}$ (with applying gate voltages) was always present, but so far, no clear explanation has been provided on this issue[15,24,26,27]. The high on/off ratio and saturation current of the FETs partially benefit from the general conductive nature of 2D materials. Novoselov et al. observed $MoS_2$ monolayer with an anomalously high doping level ($n_{2D} = 10^{12}$–$10^{13}\,cm^{-2}$) as the semi-metallic $NbSe_2$[29]. Although the electronic structures of the monolayer and bulk materials are different, the chemical activity of the surface should be similar because of the identical characteristics of the crystal structure of each layer of the van der Waals crystal. Furthermore, because the diffusion lengths of the electron injection from the surface into the bulk are ~9−16 nm (see Supplementary Note 3 for the calculation), A substantial SEA effect on the electron concentration in the monolayers, few layers, and even multilayers can be expected. According to our STM results, the estimated $n_{2D}$ values of the surface electron concentration for the non-fresh surface are in the range of $3.8 \times 10^{12}$–$6.4 \times 10^{12}\,cm^{-2}$. This range is consistent with the value of the residual $n_{2D}$ ($5.6 \times 10^{12}\,cm^{-2}$) of monolayer $MoS_2$ FETs at $V_g = 0$ reported by Kis' group[15]. The observed SEA provides a probable explanation for frequent observations of an anomalously high conductive characteristic in this 2D TMD material.

Most unintentionally doped semiconductors exhibit n-type conduction. These intrinsic n-type semiconductors usually exhibit surface electron depletion owing to the acceptor-like surface states located in the bandgap. To date, very few n-type semiconductor materials, including InAs[39–42], InN[43,44], CdO[45,46], and $In_2O_3$[47], have been found to exhibit electron accumulation rather than electron depletion on the surface. The typical SEA phenomenon is attributable to donor-like or resonance surface states located at, near, or even above the conduction band minimum. The neutral surface states at energy positions higher than the conduction band edge can easily inject electrons into the bulk region without thermal assistance (Fig. 10). This mechanism results in electron accumulation at the surface. For conventional bulk crystals, such as InAs, InN, CdO, and $In_2O_3$, the SEA characteristics have been confirmed to be inherent, such that donor-like surface states are always present, even on a clean surface. This intrinsic SEA in bulk crystals is inevitable and somewhat different from the case in the $MoS_2$ layer crystal. The

intrinsic $MoS_2$ is possible to be achieved by certain surface treatment or protective measures.

In summary, 2D electronic transport induced by SEA in $MoS_2$ nanoflakes and in the bulk was demonstrated. The surface-controlled characteristic results in a substantial thickness-dependent conductivity. The electron concentration on the surface is nearly four orders of magnitude higher than that in the inner bulk. Notably, the metal-like surface is preventable, and the intrinsic surface can be easily obtained by creating a fresh surface. The origin of the SEA has also been attributed to the formation of sulfur vacancies in the surface of $MoS_2$ due to a slow desulfurization process. This finding provides new insight into the fundamental properties of TMD layer materials and is crucial for controlling the conduction type and doping level of $MoS_2$ and for 2D device development for ultrathin flexible transparent electronics.

## Methods

**$MoS_2$ crystal growth and device fabrication**. $MoS_2$ single crystals were grown using the CVT method with bromine (Br) as a transport agent (TA). The source and crystallization ends were controlled at 1050 and 960 °C, respectively. Partial details of the crystal growth have been provided in our previous study[28]. The CVT-grown layer crystals with typical area sizes ranging from square millimeters to square centimeters were mechanically exfoliated using dicing tape to create nanoflakes with areas on the micrometer scale ($1 \times 1 - 10 \times 10\ \mu m^2$) and thicknesses ranging from a few nanometers to hundreds of nanometers. Individual nanoflakes were dispersed on the insulating $SiO_2$ (300 nm)/$n^+$-Si substrate chip with a pre-patterned Ti (30 nm)/Au (90 nm) circuit layout (see Fig. 1d, inset) prior to electrode fabrication with the nanoflakes. Two electrical contacts were made on individual $MoS_2$ nanoflakes by using dual-gun FIB (FEI Quanta 3D FEG) deposition with Pt ($100 - 500$ nm) as the contact metal. The voltage and current of the ion beam for the Pt precursor decomposition were operated at 30 kV and 100 pA, respectively. A silver paste was used for the electrical contacts on the millimeter-sized bulk crystals.

**Structural and electrical characterization of $MoS_2$ bulks and nanoflakes**. The morphological and structural properties of the $MoS_2$ layer crystals were characterized using SEM (Hitachi S-3000H), XRD (Bruker D2 Phaser), and Raman spectroscopy (Renishaw inVia Raman microscope system). The electric properties of $MoS_2$ nanoflakes were characterized using two-probe $I-V$ measurements. The thickness of the nanoflakes was measured using AFM (Bruker Dimension Icon). The electric characterization and its temperature dependence were examined on an ultralow current leakage cryogenic probe station (LakeShore Cryotronics TTP4). A semiconductor characterization system (Keithley 4200-SCS) equipped with two independent source-measure units was used to source the DC voltage and measure the current. The metal contacts of the Pt/$MoS_2$ interface of the devices were characterized using cross-sectional TEM (FEI Tecnai G2 F-20), SAED, and EDX.

**STM and STS measurements of $MoS_2$ crystals**. The STM measurements were performed under ultra-high vacuum conditions with a base pressure of ~$5 \times 10^{-11}$ Torr. To investigate the local density of states at the $MoS_2$ surface, the STS technique was utilized to obtain the $dI/dV$ as a function of the sample bias at room temperature; thus, the local topography and electronic structures of the $MoS_2$ surface could be directly observed. Prior to the STS measurement, the ex situ fresh surface of $MoS_2$ was exposed to the air for less than 30 min during preparation before being loaded into the vacuum chamber. Then, the samples stayed in vacuum for 3–4 h until the pressure reached a steady state.

**ARPES measurement of $MoS_2$ crystals**. ARPES spectra were measured at the National Synchrotron Radiation Research Center (NSRRC) in Hsinchu, Taiwan at beamline BL21B1 U9-CGM. The photoemission spectra were measured in a UHV chamber equipped with a hemispherical analyzer (Scienta R4000, collecting angle ±15°). The polarization vector was invariably in the angular dispersive plane. The $MoS_2$ single crystals with pristine, ex situ, and in situ-cleaved surfaces were measured at a base pressure of $5.6 \times 10^{-11}$ Torr. All spectra were recorded for samples at 85 K and with a 42-eV photon energy. The angular resolution was 0.2°; the energy resolution was better than 23 meV. The aging effect for the cleaved surface at 85 K and RT was stayed and recorded at a $5.6 \times 10^{-11}$ and $6.1 \times 10^{-11}$ base pressure, respectively. Prior to the ARPES measurement, the ex situ fresh surface of $MoS_2$ was exposed to the air for less than 10 min during preparation before being loaded into the vacuum chamber. Then, the samples were left in vacuum for ~1 h until the pressure ($5.6 \times 10^{-11}$ Torr) and temperature (85 K) reached the steady state. The in situ fresh surface was cleaved in a UHV atmosphere ($9 \times 10^{-11}$ Torr) at 85 K without air exposure and was subsequently measured after 5 min exposure in vacuum.

**Data availability**. The data that support the findings of this study are available from the corresponding author on request.

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

## Acknowledgements

This paper is in memory of Prof. Y.S. Huang at Taiwan Tech for his support of the MoS$_2$ bulk crystals. R.S.C. and C.-M.C. thank the support of the Ministry of Science and Technology (MOST), Taiwan under the projects MOST 105-2112-M-011-001-MY3, MOST 104-2923-M-011-001-MY3, and 105-2112-M-213-006-MY2.

## Author contributions

R.S.C. supervised the project, designed the experiments, proposed the model, and wrote the manuscript. M.D.S. and W.C.S. fabricated the devices and performed the electrical characterization. Z.W.C., M.C.S., and Y.P.C. performed the STM characterization. C.M. C. performed the ARPES characterization. All authors discussed the results and approved the final version of the manuscript.

## Additional information

**Competing interests:** The authors declare no competing interests.

