## [Peer Review File · Nature Communications]

Reviewers' comments:

Reviewer #1 (Remarks to the Author):

The main claim of this manuscript is that the surface of exposed (i.e., un-encapsulated) crystalline (CVT-grown) MoS₂ accumulates electrons due to the formation of sulfur vacancies. As a consequence, the exposed surface dominates transport in a multilayer MoS₂ device. A combination of I-V characteristics of CVT-grown multilayer MoS₂ devices (FET and TLM), STM and ARPES support this claim. The conductance fluctuates around the same value for different MoS₂ crystal thicknesses because due to the dominance of the surface accumulation layer. Devices were fabricated using a FIB technique with in-situ metal deposition resulting in Ohmic output curves (Fig. 2) and low contact resistances (Fig. 5) that are comparable with the lowest reported values so far. STS (Fig. 7) shows that the Fermi level of air exposed MoS₂ is near the conduction band edge whereas it is near midgap for freshly exfoliated MoS₂. Convincingly, the electron density estimation from the measured Fermi level position (relative to ECB) is within the range of previously reported carrier densities for MoS₂. The measured valence band edge by ARPES of in-situ cleaved MoS₂ shows a smaller tail and suggests a Fermi level closer to midgap compared to air-exposed MoS₂ (Fig. 8), in qualitative agreement with STS outcome. This work provides novel and important insights as it probes the factors (by multiple complimentary methods) that affect the band structure of a 2D semiconductor surface, and its importance for 2D device transport. Nonetheless, some inconsistencies and additional points need to be addressed.

1. For thick devices, the gate field is screened by the accumulation layer and the field effect mobility values are not accurate. The assumptions used to derive equation (5) do not hold.
2. To what thickness does the surface accumulation layer persist? Most devices are one to a few layers thick. The underlying dielectric may have a substantial effect, which is here neglected because the devices are so thick. The implications for most devices is not so clear.
3. The effect of ion beam exposure on the surface of the exposed crystals has not been considered. There will be substantial proximity dosing, and hence damage, due to this procedure.
4. The exposure time and condition is critical as it determines the electronic properties of the exposed surface in multilayer MoS₂: 30 minutes of air-exposure resulted in MoS₂ with a Fermi level close to midgap suggesting intrinsic properties whereas only 10 min of exposure to air induced an energy shift of 0.31 eV according to ARPES (attributed to a Fermi level). These results seem to be conflicting therefore further clarification is required. What role does ambient air play for the formation of sulfur vacancies?
5. ARPES measurements of MoS₂ that was kept for 11 h in UHV show a valence band shift, even at low temperatures (110 °C). This is attributed to the formation of sulfur vacancies which should increase the carrier density in the outermost MoS₂ layer. If this is the case, then the band alignment of MoS₂ measured by STS at RT is expected to change with time, and should qualitatively agree with the energy shift observed with ARPES.
6. What conditions (time of air exposure, annealing at moderate temperatures in vacuum) start to affect the transport properties of the MoS₂ devices? For instance, ARPES of annealed MoS₂ (110°C for 20 min in UHV) suggest a dramatic effect on its electronic structure. However, these annealing conditions are considered mild when compared with standard fabrication procedures of MoS₂ devices. The incorrect field effect mobility measurements (see point 1) complicate comparison with the literature.
7. A comparison of the measured E_g with previous STS reports is missing.

Reviewer #2 (Remarks to the Author):

Siao et al. present a systematic study of the surface properties of MoS₂ flakes by thickness-dependent transport, STM, and ARPES. Their results are broadly consistent with surface electron accumulation originating from some changes of the surface over time. This doesn't seem very surprising to me. The Fermi level will always be pinned at the surface by a density of defects. For the intrinsic bulk, this will give rise to a small electron accumulation or depletion depending on where the Fermi defect states are, but all semiconductors have this effect. It is not very strong here - the conduction band states are not significantly populated, no quantum well states are observed as in the other examples cited here. The situation is not like the schematic shown in Figure 10 where a much stronger effect is shown.

Loss of sulfur at the surface seems plausible, but there is no real evidence for this (can they observe this by the density of sulfur vacancies as seen by STM in the different samples, for example?).

It is of value for the community to know of these time-dependent changes, and so this work should be published, but to me I am afraid I do not see the novelty that would suggest it should be published in Nature Communications.

Reviewer #3 (Remarks to the Author):

The claim made that a Surface Electron Accumulation (SEA) layer occurs in MoS₂ is both novel and of great importance to the 2D electronics community. The experiments used to draw this conclusion were well thought-out/executed and resulted in a data set that supported the claims made. The central point that all experiments (on nanoflakes and bulk) were made on materials from the same growth removes (or at least reduces) a major source of variability.

The direct observation of the desulfurization via atomic-level imaging would be even more convincing. However, I agree that the ARPES data supports conclusion that the SEA is most likely not induced by foreign molecules such as oxygen or water.

To Reviewer #1:

We sincerely thank the reviewer's comments and suggestions to help improve the manuscript quality. We detail below our responses to the comments and suggestions raised by the reviewer. All the changes in the manuscript have been highlighted in yellow.

COMMENTS: The main claim of this manuscript is that the surface of exposed (i.e., un-encapsulated) crystalline (CVT-grown) MoS₂ accumulates electrons due to the formation of sulfur vacancies. As a consequence, the exposed surface dominates transport in a multilayer MoS₂ device. A combination of I-V characteristics of CVT-grown multilayer MoS₂ devices (FET and TLM), STM and ARPES support this claim. The conductance fluctuates around the same value for different MoS₂ crystal thicknesses because due to the dominance of the surface accumulation layer. Devices were fabricated using a FIB technique with in-situ metal deposition resulting in Ohmic output curves (Fig. 2) and low contact resistances (Fig. 5) that are comparable with the lowest reported values so far. STS (Fig. 7) shows that the Fermi level of air exposed MoS₂ is near the conduction band edge whereas it is near midgap for freshly exfoliated MoS₂. Convincingly, the electron density estimation from the measured Fermi level position (relative to ECB) is within the range of previously reported carrier densities for MoS₂. The measured valence band edge by ARPES of in-situ cleaved MoS₂ shows a smaller tail and suggests a Fermi level closer to midgap compared to air-exposed MoS₂ (Fig. 8), in qualitative agreement with STS outcome. This work provides novel and important insights as it probes the factors (by multiple complimentary methods) that affect the band structure of a 2D semiconductor surface, and its importance for 2D device transport. Nonetheless, some inconsistencies and additional points need to be addressed.

Question 1: For thick devices, the gate field is screened by the accumulation layer and the field effect mobility values are not accurate. The assumptions used to derive equation (5) do not hold.

Response 1: Equation (5) is based on the operation principle of MOSFET. For *n*-type semiconductors, a lower electron concentration (*n*) makes a more sensitive change of

depletion region in semiconductor and thus higher dI_{ds}/dV_g under the gate field variation. The mobility value is derived from the n and I_{ds} (or conductance) values. Accordingly, the gate field effect on the depletion region is the key mechanism determining whether Equation (5) is applicable in this case.

Indeed, different from the general unintentionally doped n -type semiconductors with a surface depletion region (SDR), MoS₂ has a surface accumulation region instead. The band alignment of the MoS₂ MOSFET in equilibrium is drawn schematically as Fig. R1(a). In this configuration, by applying the forward or reverse gate voltage, we can expect that the accumulation region changes with gate voltage. At forward gate voltage, electron accumulation increases near the interface of gate oxide and MoS₂ and the source-drain conductance of the conducting channel also increases, Fig. R1(b). On the other hand, at reverse gate voltage, electron accumulation decreases and the conductance also decreases, Fig. R1(c).

Because the accumulation layer originates from the donor-like surface states, it makes an influence like heavy doping in the semiconductor. The only difference is that this kind of doping is only limited in the near surface area rather than homogeneous doping in the bulk. However, operation of MOSFET is also limited in the region close to gate oxide. This means that the gate field is still possible to influence the accumulation layer in MoS₂. Now the situation is more like the MOSFET with heavily doped MoS₂ and we can treat MoS₂ with a naturally occurred n^{++} - n junction at the region near the surface. Accordingly, the mobility value defined by the FET measurement for the pristine (nonfresh) MoS₂ can be considered as the transport property of the accumulated surface electrons. Also due to the conductance of the pristine MoS₂ is dominated by the accumulation layer, we can attribute the FET mobility of the accumulation layer to the inherent transport property in the nonfresh MoS₂. The aforementioned statement probably can justify our comparison to the mobility values of the fresh and nonfresh MoS₂ flakes.

In addition, the electron concentration of the accumulation layer is at the orders of magnitude of 10^{19} cm^{-3} . Usually to make a substantial screen effect, the electron densities up to 10^{22} - 10^{24} cm^{-3} , which are the general levels of metals, are required. The electron concentration at 10^{19} cm^{-3} still belongs to the value of heavily doped semiconductors. Actually the similar MOSFET measurement has also been demonstrated for the indium nitride (InN) nanowires by Mark Reed's group.^[Ref. R1] InN, which is a well known low-bandgap semiconductor, also possesses electron surface accumulation. Its surface

electron density is as high as 10^{21} cm^{-3} .^[Ref. R2] According to their report, the field effect can still be observed for the material with even more substantial electron accumulation.

The comparison of the field-effect mobility between the fresh and nonfresh samples indicates a substantial surface scattering in MoS₂. The statement is somewhat consistent with the previous report.^[Ref. R3] Lu *et al* reported the thickness-dependent mobilities for the MoS₂ nanoflakes (1 to 15 layers). The scattering from surface defects was proposed to be a dominant factor determining the monotonic increase of mobility with layer number. The information has been added in the revised manuscript, Page 20, lines 5-9, which might be able to further support our conclusion.

Ref. R1. Cheng, G., Stern, E., Turner-Evans, D., Reed, M. A. Electronic properties of InN nanowires. *Appl. Phys. Lett.* **87**, 253103 (3 pages) (2005).

Ref. R2. Mahboob, I., Veal, T. D., McConville, C. F., Lu, H., Schaff, W. J. Intrinsic electron accumulation at clean InN surfaces. *Phys. Rev. Lett.* **92**, 036804 (4 pages) (2004).

Ref. R3. Lu, C. P., Li, G., Mao, J., Wang, L. M., Andrei, E. Y. Bandgap, mid-Gap states, and gating effects in MoS₂. *Nano Lett.* **14**, 4628-4633 (2014).

Figure R1. Schematic band alignment of the MOSFET based on the MoS₂ with surface electron accumulation (a) in equilibrium ($V_g = 0$), (b) at forward gate voltage ($V_g > 0$), and (c) at reverse gate voltage ($V_g < 0$). n^+ -Si/MoS₂ is treated as the metal/semiconductor, E_F^m/E_F^s is the Fermi level of metal/semiconductor, and E_C^s is the conduction band minimum of semiconductor.

Question 2: To what thickness does the surface accumulation layer persist? Most devices are one to a few layers thick. The underlying dielectric may have a substantial effect, which is here neglected because the devices are so thick. The implications for most devices is not so clear.

Response 2: The surface electron accumulation originates from the electron injection from the surface (donor-like surface states) into the bulk. The surface-bulk interface is somewhat similar to the n^{++} - n junction. To understand to what thickness the electron accumulation persists, we can estimate the diffusion length of electron (L_n) in the intrinsically n-type MoS₂. According to the one-dimensional continuity equation at steady

state: $D_n \frac{d^2(\delta n)}{dx^2} - \frac{\delta n}{\tau_n}$, the excess electron concentration (δn) as a function of space (x) is written as $\delta n(x) = \delta n(0) \exp(-x/L_n)$, where $\delta n(0)$ is the excess electron concentration at $x = 0$ and $L_n = (D_n \tau_n)^{1/2}$, where D_n is the diffusion coefficient of electron and τ_n (~1 ns)^[Ref. R4] is the electron lifetime.^[Ref. R5] The D_n of MoS₂ can be calculated by the Einstein equation, $D_n / \mu_n = kT/q$, where μ_n is the electron mobility, k is Boltzmann's constant, T is the temperature set at 300 K, and q is the elementary charge.

The layer material like MoS₂ has strong anisotropic transport properties. The conductivity and effective mass of electron perpendicular to the c -axis (in-plane) is approximately three orders of magnitude higher than that along c -axis (out-of-plane).^[Ref. R6] According to the references, the in-plane mobility values of MoS₂ bulk are 32–100 cm²V⁻¹s⁻¹^[Ref. R7,R8] and so the out-of-plane mobility values are inferred at the range of 0.032–0.1 cm²V⁻¹s⁻¹. Because the electron injection is along the c -axis, the out-of-plane mobilities were adopted for the D_n calculation. The estimated D_n are 8.3×10^{-4} – 2.6×10^{-3} cm²s⁻¹ and the obtained L_n locates in the range of 9.1–16 nm.

The flake thicknesses of our devices are in the range of 30–400 nm. For the monolayer and few-layer devices, the thicknesses (0.7–5 nm) are much lower than the diffusion length, which implies that most devices suffer significant doping by the surface and become more conductive. This statement can be supported by the values estimated from the literatures. Actually an unusually high 2D electron concentration (n_{2D}) range at 2×10^{13} – 6.5×10^{15} cm⁻² has been reported for the MoS₂ FETs (with applying gate voltages) but so far no clear explanation on this issue.^[Ref. R9-R12] Applying gate field could make a high n_{2D} at accumulation mode artificially. To rule out the influence of gate field, Kis's group reported an residual 2D carrier concentration as high as 5.6×10^{12} cm⁻² at zero gate voltage by the Hall measurement.^[Ref. R10] If we convert the n_{2D} values to their 3D values, we will get the electron concentration at 1.3×10^{19} cm⁻³. The values estimated from the reference are quite consistent with our STM measurement ($n \sim 10^{19}$ cm⁻³).

The analysis suggests that the anomalously high conductive nature in the most MoS₂ devices probably originates from the SEA. Also due to the high electron concentration, when the MoS₂ FETs were depleted by the reverse gate voltage, the conductance difference between the on-state (accumulation mode at zero and forward bias) and off-state (depletion mode at reverse bias) can reach to 10⁸ and 10⁹.

The aforementioned discussion can be found in the revised manuscript, Page 17, lines 1-9 from the bottom and Page 18, lines 1-9. The calculation of diffusion length has also been added in the Supplementary Information.

Ref. R4. Wang, H., Zhang, C., Rana, F. Surface Recombination Limited Lifetimes of Photoexcited Carriers in Few-Layer Transition Metal Dichalcogenide MoS₂. *Nano Lett.* **15**, 8204-8210 (2015).

Ref. R5. Neamen, D. A. *Semiconductor Physics and Devices: Basic Principles*, 3rd edition (McGraw-Hill Inc., New York, **2003**), Chap. 6, pp. 206-207.

Ref. R6. Guha Thakurta, S. R., Dutta, A. K. Electrical conductivity, thermoelectric power and Hall effect in *p*-type molybdenite (MoS₂) crystal. *J. Phys. Chem. Solids* **44**, 407-416 (1983).

Ref. R7. Tiong, K. K., Liao, P. C., Ho, C. H., Huang, Y. S. Growth and characterization of rhenium-doped MoS₂ single crystals. *J. Crystal Growth* **205**, 543-547 (1999).

Ref. R8. Fivaz, R., Mooser, E. Mobility of charge carriers in semiconducting layer structures. *Phys. Rev.* **163**, 743-755 (1967).

Ref. R9. Bertolazzi, S., Krasnozhan, D., Kis, A. Nonvolatile memory cells based on MoS₂/Graphene heterostructures. *ACS Nano* **7**, 3246-3252 (2013).

Ref. R10. Radisavljevic, B., Kis, A. Mobility engineering and a metal-insulator transition in monolayer MoS₂. *Nature Mater.* **12**, 815-820 (2013).

Ref. R11. Zhang, Y., Ye, J., Matsushashi, Y., Iwasa, Y. Ambipolar MoS₂ thin flake transistors. *Nano Lett.* **12**, 1136-1140 (2012).

Ref. R12. Park, W., Park, J., Jang, J., Lee, H., Jeong, H., Cho, K., Hong, S., Lee, T. Oxygen environmental and passivation effects on molybdenum disulfide field effect transistors. *Nanotechnology* **24**, 095202 (5 pages) (2013).

Question 3: The effect of ion beam exposure on the surface of the exposed crystals has not been considered. There will be substantial proximity dosing, and hence damage, due to this procedure.

Response 3: Actually, we also had the concern that the electrode fabrication under

ion-beam environment could create the extra surface defects leading to the “artificial” high surface conductivity. So according to our earlier study (unpublished), we tried to minimize the ion beam exposure time when conducting the Pt deposition. Usually we made everything ready at SEM mode and then switched to FIB mode. The whole sample surface was only exposed to the ion beam at a very short time (by snap shot function) when the FIB image was needed for defining Pt deposition areas.

Furthermore, to make a more convincing conclusion, we also made the surface protection by coating the insulating organic material (bathocuproine, BCP) on the MoS₂ prior to FIB fabrication. Figure R2 depicts the FESEM images, AFM profile, and IV curve for a typical nonfresh flake device with BCP protection. The statistic conductivity values for three nonfresh MoS₂ flakes with BCP protection are illustrated in Figure R3. The result shows that the conductivities of the BCP-coated nonfresh flakes (red star) are very close to the values of the samples without BCP protection (blue sphere). Considering the BCP layer needs to be removed for the BCP-coated devices before Pt deposition, the process could produce an extra contact resistance if the BCP layer was not entirely removed. Accordingly, the average conductivity of the BCP-coated MoS₂ slightly lower than that of the samples without BCP protection is difficult to conclude that the FIB processing generates extra surface states or doping in the nonfresh MoS₂ flakes.

In addition, when comparing the fresh and nonfresh samples all with surface protection, we can observe that the BCP-coated nonfresh flakes are much more conductive than the SiO₂-coated fresh flakes (green square). The result indicates that even without the influence of ion beam bombardment, the surface-protected nonfresh flakes have the high conductivities in nature. The ion beam exposure might be treated as a minor effect on the conductivity of the nonfresh (pristine) MoS₂ flakes.

Figure R2. (a) Top-viewed and (b) perspective-viewed FESEM images, (c) AFM profile, and (d) IV measurement for a typical FIB-fabricated nonfresh MoS₂ flake device with BCP surface coating.

Figure R3. Thickness-dependent conductivity plots for the nonfresh MoS₂ flakes with and without BCP protection and for the fresh flakes with SiO₂ protection.

Question 4: The exposure time and condition is critical as it determines the electronic

properties of the exposed surface in multilayer MoS₂: 30 minutes of air-exposure resulted in MoS₂ with a Fermi level close to midgap suggesting intrinsic properties whereas only 10 min of exposure to air induced an energy shift of 0.31 eV according to ARPES (attributed to a Fermi level). These results seem to be conflicting therefore further clarification is required. What role does ambient air play for the formation of sulfur vacancies?

Response 4: Definition of the position of conduction band minimum (CBM) and valance band maximum (VBM) by STS spectra is not very precise due to the tip-induced band bending (TIBB) effect, especially for insulating surface (like the ex-situ fresh MoS₂ surface), so the bandgap (E_g) in Fig. 7(g) is overestimated at 1.84-1.90 eV (theoretical $E_g \approx 1.3$ eV at room temperature). Actually according to Fig. 7(g), the Fermi level shows an observable blue-shift (0.24–0.33 eV) to CBM from the midgap position.

In addition, according to ARPES results, the VBM can be determined from the normal emission spectrum by probed band structure directly. The Fermi level or VBM shift is 0.23 eV between the in-situ fresh (VBM at -0.75 eV) and ex-situ fresh surfaces (10 min air exposure, VBM at -0.98 eV) in Fig. 8(e). Because ARPES measurement was conducted at low temperature (85 K) and STS was measured at room temperature (RT), energy shift can be magnified due to bandgap increase at low-T ARPES measurement. Also, unlike ARPES method, the energy shift defined by PDOS from STS measurement could have even higher uncertainty due to the TIBB effect at the insulating surface. Comparison of the values of energy shift between ARPES and STS is kind of difficult in our case. Based on these reasons, we did not try to make a quantitative comparison in the energy shift values between the ARPES and STS measurements.

The effect of ambient air such as oxygen and water molecules indeed need further investigation. At this stage, we can only make sure that the sulfur vacancy formation is spontaneous and occurs even at vacuum and at low temperature (according to ARPES result, Fig. 9). It looks like that the assistance of oxygen or water molecules is not necessary. However, we can not totally rule out the effect of ambient molecules. Actually it is possible that the interaction between the MoS₂ surface and oxygen/water molecule could speed up the desulphurization rate and enhance the formation of sulfur vacancy in air ambience. However, it is kind of difficult to clarify the ambient air effect because the ARPES measurements have to be conducted in ultra-high vacuum. A special design is required to inject oxygen or water gas into the vacuum chamber to monitor the change of ARPES

spectra. Further investigation is still required to elaborate this point in the future.

Question 5: ARPES measurements of MoS₂ that was kept for 11 h in UHV show a valence band shift, even at low temperatures (110 °C). This is attributed to the formation of sulfur vacancies which should increase the carrier density in the outermost MoS₂ layer. If this is the case, then the band alignment of MoS₂ measured by STS at RT is expected to change with time, and should qualitatively agree with the energy shift observed with ARPES.

Response 5: Because of the different measurement conditions (Low-T and RT) and operation principles (TIBB effect in STS measurement), the energy shift values defined by the ARPES and STS have different uncertainties and are difficult to be compared quantitatively. Please see our response to Q4 for the details. However, qualitatively, we can still observe the consistent blue-shift of Fermi level for the two different approaches. For the STS, the Fermi level moves substantially to near CBM after long-term air exposure, Fig. 7. For the ARPES, the longer exposure time in air (< 10 min) and in vacuum (~1 h) for the ex-situ fresh surface also gives rise to more blue-shift of the Fermi level compared to the in-situ fresh surface, Fig. 8.

Though there exist the fundamental differences between the STS and ARPES approaches, to help readers understand the detailed histories of the studied surfaces since the surfaces being created, the information of exposure time either in air or in vacuum has been added in the revised manuscript, Methods portion, Page 22, lines 5-8 from the bottom and Page 23, lines 7-12.

Question 6: What conditions (time of air exposure, annealing at moderate temperatures in vacuum) start to affect the transport properties of the MoS₂ devices? For instance, ARPES of annealed MoS₂ (110°C for 20 min in UHV) suggest a dramatic effect on its electronic structure. However, these annealing conditions are considered mild when compared with standard fabrication procedures of MoS₂ devices. The incorrect field effect mobility measurements (see point 1) complicate comparison with the literature.

Response 6: Because the electron accumulation increases with time, theoretically transport properties (or conductivity) are governed gradually by the surface. According to

the pristine sample in Fig. 8(e) and the annealed sample in Fig. 9(e), the Fermi level position of the annealed surface is higher than that of the pristine surface. This result suggests that the annealing treatment for only 20 min in UHV produces a more significant effect than the long-term exposure at RT. We can expect that the electron concentration or conductivity can be further increased after annealing treatment for the pristine devices.

We ever tried to measure the long-term conductivity versus exposure time curve to obtain the information of the threshold exposure time starting to affect the conductivity. However, the result is not conclusive. The major reason is that the ohmic contact is difficult to be achieved on the fresh MoS₂ surface (using bulk samples). Though FIB provides better ohmic contacts, the fresh nanoflakes need SiO₂ coating to prevent potential electron beam and ion beam damage. So, FIB-fabricated devices can be not used to monitor the effect of air exposure time. Without the stable electric contacts on the bulk MoS₂ with the fresh surface, the current fluctuation is too high to achieve a convincing result.

In addition, it always takes a few hours to fabricate a bulk device and the fresh surface is inevitably exposed in air during the process. This short-term air exposure which is capable to make a preliminary Fermi level shift has been confirmed by ARPES. This means the conductivity versus exposure time measurement is only available for the ex-situ fresh surface, which also increases the difficulty to observe the time-dependent transport properties.

Regarding the issue of incorrect field-effect mobility, we have detailed our explanation in the response to Q1, which might be able to justify the comparison of mobility.

Question 7: A comparison of the measured E_g with previous STS reports is missing.

Response 7: The bandgap determined by our STS measurement for the nonfresh MoS₂ is approximately 1.35 ± 0.05 eV. Due to the strong tip-induced band-bending (TIBB) effect in the fresh MoS₂ surface which is much more insulating, the bandgap is overestimated at 1.8–1.9 eV. Though there are quite a few STS studies on MoS₂, most of them concentrated on the monolayer structure rather than bulk MoS₂. Only one reference regarding the study in the bandgap value on the MoS₂ bulk crystals using STS was found. Lu *et al* reported a statistic result of bandgap value at 1.29 ± 0.045 eV for bulk MoS₂,^[Ref. R3] which is consistent with that of our measurement (The lower bound of our measured bandgap value at 1.28

eV is consistent with the mean value in the Ref. R3). The comparison of the bandgap measured by STS has been briefly discussed and added in the revised manuscript, Page 13, lines 1-4 from the bottom and Page 14, lines 1-2.

Ref. R3. Lu, C. P., Li, G., Mao, J., Wang, L. M., Andrei, E. Y. Bandgap, mid-Gap states, and gating effects in MoS₂. *Nano Lett.* **14**, 4628-4633 (2014).

To Reviewer #2:

We sincerely thank the reviewer's comments and suggestions to help improve the manuscript quality. We detail below our responses to the comments and suggestions raised by the reviewer. All the changes in the manuscript have been highlighted in yellow.

COMMENTS: Siao et al. present a systematic study of the surface properties of MoS₂ flakes by thickness-dependent transport, STM, and ARPES. Their results are broadly consistent with surface electron accumulation originating from some changes of the surface over time. This doesn't seem very surprising to me. The Fermi level will always be pinned at the surface by a density of defects. For the intrinsic bulk, this will give rise to a small electron accumulation or depletion depending on where the Fermi defect states are, but all semiconductors have this effect. It is not very strong here - the conduction band states are not significantly populated, no quantum well states are observed as in the other examples cited here. The situation is not like the schematic shown in Figure 10 where a much stronger effect is shown.

It is of value for the community to know of these time-dependent changes, and so this work should be published, but to me I am afraid I do not see the novelty that would suggest it should be published in Nature Communications.

Response 1: Regarding the degree of electron accumulation, probably we can compare MoS₂ with other electron accumulation system to address this point. If simply comparing the semiconductors (InAs, InN, In₂O₃ and CdO) which have been found to possess surface electron accumulation (SEA) properties, the surface electron concentration (n_s) values are $8 \times 10^{20} \text{ cm}^{-3}$ for CdO,^[Ref. R1] $2.8 \times 10^{20} \text{ cm}^{-3}$ for InN,^[Ref. R2] $1.9 \times 10^{19} \text{ cm}^{-3}$ for In₂O₃,^[Ref. R3] and $1.6\text{--}10 \times 10^{17} \text{ cm}^{-3}$ for InAs.^[Ref. R4,R5] MoS₂ has the n_s at $\sim 1 \times 10^{19} \text{ cm}^{-2}$, which is much higher than that of InAs and is comparable with that of In₂O₃.

In addition, considering the electron concentration difference between the surface and the inner bulk (n_b), the ratios of n_s/n_b for the known SEA systems are in the range of 2.6–50.^[Ref. R1-R5] However, these values are much lower than that ($>10^3$) of MoS₂. The higher n_s/n_b ratio stands for that the MoS₂ could exhibit more significant SEA-induced effects than the other electron accumulation systems. This statement can explain why the

thickness effects on the transport properties are very significant in MoS₂ but were not easy to be observed in the other materials with SEA. The comparison probably can provide the information of which MoS₂ could possess a strong SEA effect compared to other electron accumulation systems.

Ref. R1. Piper, L. F. J. et al. Observation of quantized subband states and evidence for surface electron accumulation in CdO from angle-resolved photoemission spectroscopy. *Phys. Rev. B* **78**, 165127 (5 pages) (2008).

Ref. R2. Mahboob, I., Veal, T. D., McConville, C. F., Lu, H., Schaff, W. J. Intrinsic electron accumulation at clean InN surfaces. *Phys. Rev. Lett.* **92**, 036804 (4 pages) (2004).

Ref. R3. King, P. D. C. et al. Surface electron accumulation and the charge neutrality level in In₂O₃. *Phys. Rev. Lett.* **101**, 11608 (4 pages) (2008).

Ref. R4. Noguchi, M., Hirakawa, K., Ikoma, T. Intrinsic electron accumulation layers on reconstructed clean InAs(100) surfaces. *Phys. Rev. Lett.* **66**, 2243-2246 (1991).

Ref. R5. Betti, M. G., Corradini, V., Bertoni, G., Casarini, P., Mariani, C., Abramo, A. Density of states of a two-dimensional electron gas at semiconductor surfaces. *Phys. Rev. B* **63**, 155315 (10 pages) (2001).

Comment 2: Loss of sulfur at the surface seems plausible, but there is no real evidence for this (can they observe this by the density of sulfur vacancies as seen by STM in the different samples, for example?).

Response 2: Over the past few months, we have been trying to take the STM images of sulfur vacancy (SV) with atomic resolution. Limited by the recent status of our STM machine, we still failed to access the atom-resolved images. However, according to our STM measurement together with STS analysis, probably we can provide an alternative evidence to show the presence of the SV-related defects in the MoS₂ surface.

Figures R1(a) and R1(b) depicts a STM image and its STS mapping targeting an area with a point defect in the fresh MoS₂ surface. By taking the STS spectra at the defect center (position D) and its nearby area (positions N1 and N2), we can clearly observe an extra peak close to the valance band (VB) edge centered at -1.4 V for the spectrum measured at

position D as shown in Fig. R1(c). The similar feature is not so significant at the positions near the defect site (N1 and N2). According to the literatures, the presence of the SV introduces the defect states in the bandgap close to the VB edge.^[Ref. R6-R9] The Fermi level at the defect center (D) exhibiting the blue-shift for 0.12–0.20 eV compared to those of the nearby positions (N1 and N2) indicates the *n*-type nature of the SV, which is also consistent with the previous reports.^[Ref. R10]

Furthermore, the same STM and STS measurements have also been conducted for the pristine (non-fresh) surface. Figures R2(a) and R2(b) depicts the STM image and its STS mapping targeting an area with two close point defects in the nonfresh MoS₂ surface. According to the STS spectra in Figs. R2(c) and R2(d), we can also observe the similar features close to VB edge at the positions of defect centers (D1 and D2). The intensity of the extra peak in the non-fresh surface is not as high as that in the fresh sample, which is probably due to the long-term exposure in air. Santosh et al suggest that the oxygen adsorption could suppress the states originating from the SV.^[Ref. R9] The energy distribution and its density of states (DOS) could be slightly changed by the interaction of SV and oxygen molecules. The increase of the SV density could make a broader distribution of the defect states.^[Ref. R7] These statements probably can explain that the SV-related features are slightly different in energy position and relative intensity in the STS spectra shown in Figs. R1(c), R2(c) and R2(d).

Because the manuscript is a bit lengthy already, the aforementioned STM and STS results are briefly mentioned in the revised manuscript, Page 16, lines 1-3 from the bottom and Page 17, lines 1-3 and have been detailed in the Supplementary Information, Figs. S2 and S3.

Ref. R6. Fuhr, J. D., Saul, A., Sofo, J. O. Scanning tunneling microscopy chemical signature of point defects on the MoS₂(0001) surface. *Phys. Rev. Lett.* **92**, 026802 (4 pages) (2004).

Ref. R7. Qiu, H., Xu, T., Wang, Z., Ren, W., Nan, H., Ni, Z., Chen, Q., Yuan, S., Miao, F., Song, F., Long, G., Shi, Y., Sun, L., Wang, J., Wang, X. Hopping transport through defect-induced localized states in molybdenum disulphide. *Nature Commun.* **4**, 2642 (6 pages) (2013).

Ref. R8. Santosh, K. C., Longo, R. C., Addou, R., Wallace, R. M., Cho, K. Impact of intrinsic atomic defects on the electronic structure of MoS₂ monolayers. *Nanotechnology* **25**, 375703 (6 pages) (2014).

Ref. R9. Akdim, B., Pachter, R., Mou S. Theoretical analysis of the combined effects of sulfur vacancies and analyte adsorption on the electronic properties of single-layer MoS₂. *Nanotechnology* **27**, 185701 (10 pages) (2016).

Ref. R10. McDonnell, S., Addou, R., Buie, C., Wallace, R. M., Hinkle, C. L. Defect-dominated doping and contact resistance in MoS₂. *ACS Nano* **8**, 2880-2888 (2014).

Figure R1. (a) STM image and (b) its STS mapping targeting an area with a point defect in the fresh MoS₂ surface. The white dash arrow in (a) indicates the scanning region of STS mapping. The red arrows in (b) indicate the influence region of the point defect labelled with D. (c) STS spectra measured at the defect center (position D) and its nearby area (positions N1 and N2) marked in (a).

Figure R2. (a) STM image and (b) its STS mapping targeting an area with two point defects in the pristine (nonfresh) MoS₂ surface. The white dash arrow in (a) indicates the scanning region of STS mapping. The red arrows in (b) indicate the influence regions of the point defects labelled with D1 and D2. (c) STS spectra measured at the defect center (position D1) and its nearby area (positions N1 and N2) marked in (a). (d) STS spectra measured at the defect center (position D2) and its nearby area (positions N3 and N4) marked in (a).

To Reviewer #3:

We sincerely thank the reviewer's comments and suggestions to help improve the manuscript quality. We detail below our responses to the comments and suggestions raised by the reviewer. All the changes in the manuscript have been highlighted in yellow.

COMMENTS: The claim made that a Surface Electron Accumulation (SEA) layer occurs in MoS₂ is both novel and of great importance to the 2D electronics community. The experiments used to draw this conclusion were well thought-out/executed and resulted in a data set that supported the claims made. The central point that all experiments (on nanoflakes and bulk) were made on materials from the same growth removes (or at least reduces) a major source of variability.

Question 1: The direct observation of the desulfurization via atomic-level imaging would be even more convincing. However, I agree that the ARPES data supports conclusion that the SEA is most likely not induced by foreign molecules such as oxygen or water.

Response 1: Over the past few months, we have been trying to take the STM images of sulfur vacancy (SV) with atomic resolution. Limited by the recent status of our STM machine, we still failed to access the atom-resolved images. However, according to our STM measurement together with STS analysis, probably we can provide an alternative evidence to show the presence of the SV-related defects in the MoS₂ surface.

Figures R1(a) and R1(b) depicts a STM image and its STS mapping targeting an area with a point defect in the fresh MoS₂ surface. By taking the STS spectra at the defect center (position D) and its nearby area (positions N1 and N2), we can clearly observe an extra peak close to the valance band (VB) edge centered at -1.4 V for the spectrum measured at position D as shown in Fig. R1(c). The similar feature is not so significant at the positions near the defect site (N1 and N2). According to the literatures, the presence of the SV introduces the defect states in the bandgap close to the VB edge.^[Ref. R1-R4] The Fermi level at the defect center (D) exhibiting the blue-shift for 0.12–0.20 eV compared to those of the nearby positions (N1 and N2) indicates the *n*-type nature of the SV, which is also consistent with the previous reports.^[Ref. R5]

Furthermore, the same STM and STS measurements have also been conducted for the pristine (non-fresh) surface. **Figures R2(a) and R2(b)** depicts the STM image and its STS mapping targeting an area with two close point defects in the nonfresh MoS₂ surface. According to the STS spectra in **Figs. R2(c) and R2(d)**, we can also observe the similar features close to VB edge at the positions of defect centers (D1 and D2). The intensity of the extra peak in the non-fresh surface is not as high as that in the fresh sample, which is probably due to the long-term exposure in air. Santosh et al suggest that the oxygen adsorption could suppress the states originating from the SV.^[Ref. R4] The energy distribution and its density of states (DOS) could be slightly changed by the interaction of SV and oxygen molecules. The increase of the SV density could make a broader distribution of the defect states.^[Ref. R2] These statements probably can explain that the SV-related features are slightly different in energy position and relative intensity in the STS spectra shown in **Figs. R1(c), R2(c) and R2(d)**.

Because the manuscript is a bit lengthy already, the aforementioned STM and STS results are briefly mentioned in the revised manuscript, **Page 16, lines 1-3 from the bottom** and **Page 17, lines 1-3** and have been detailed in the Supplementary Information, **Figs. S2 and S3**.

Ref. R1. Fuhr, J. D., Saul, A., Sofo, J. O. Scanning tunneling microscopy chemical signature of point defects on the MoS₂(0001) surface. *Phys. Rev. Lett.* **92**, 026802 (4 pages) (2004).

Ref. R2. Qiu, H., Xu, T., Wang, Z., Ren, W., Nan, H., Ni, Z., Chen, Q., Yuan, S., Miao, F., Song, F., Long, G., Shi, Y., Sun, L., Wang, J., Wang, X. Hopping transport through defect-induced localized states in molybdenum disulphide. *Nature Commun.* **4**, 2642 (6 pages) (2013).

Ref. R3. Santosh, K. C., Longo, R. C., Addou, R., Wallace, R. M., Cho, K. Impact of intrinsic atomic defects on the electronic structure of MoS₂ monolayers. *Nanotechnology* **25**, 375703 (6 pages) (2014).

Ref. R4. Akdim, B., Pachter, R., Mou S. Theoretical analysis of the combined effects of sulfur vacancies and analyte adsorption on the electronic properties of single-layer MoS₂. *Nanotechnology* **27**, 185701 (10 pages) (2016).

Ref. R5. McDonnell, S., Addou, R., Buie, C., Wallace, R. M., Hinkle, C. L. Defect-dominated doping and contact resistance in MoS₂. *ACS Nano* **8**, 2880-2888 (2014).

Figure R1. (a) STM image and (b) its STS mapping targeting an area with a point defect in the fresh MoS₂ surface. The white dash arrow in (a) indicates the scanning region of STS mapping. The red arrows in (b) indicate the influence region of the point defect labelled with D. (c) STS spectra measured at the defect center (position D) and its nearby area (positions N1 and N2) marked in (a).

Figure R2. (a) STM image and (b) its STS mapping targeting an area with two point defects in the pristine (nonfresh) MoS₂ surface. The white dash arrow in (a) indicates the scanning region of STS mapping. The red arrows in (b) indicate the influence regions of the point defects labelled with D1 and D2. (c) STS spectra measured at the defect center (position D1) and its nearby area (positions N1 and N2) marked in (a). (d) STS spectra measured at the defect center (position D2) and its nearby area (positions N3 and N4) marked in (a).

REVIEWERS' COMMENTS:

Reviewer #1 (Remarks to the Author):

The rebuttal incompletely addresses my concern about the correctness of the field effect mobility expression used in this context. That said, most authors use the same expression for field effect mobility, and the comparison will be useful though not surprising (i.e. a higher mobility away from the interface). I note that Figure R1 assumes that there is only an accumulation layer adjacent the SiO₂ above the back-gate, and not on the exposed or encapsulated surface of the channel. However, the most direct data (ARPES, STS) address the exposed interface, not the buried interface. The manuscript must be more clear in its claims regarding where a SEA exists (at a single interface or two interfaces), and how this affects the interpretation of the transport measurements. This can be addressed by a brief addition to the discussion.

The rebuttal does not establish the influence of the FIB preparation procedure on the MoS₂ properties, but it does show that the difference in fresh and non-fresh flakes is observable despite whatever influence there is.

Reviewer #2 (Remarks to the Author):

With the addition of spatially-resolved STS measurement the paper has been improved by adding the microscopic insight. In line with the comments of the other referee I am happy to recommend the paper is published in Nature Communications.

Reviewer #3 (Remarks to the Author):

In my opinion, the points raised in the previous round of review have been satisfactorily addressed.

To Reviewer #1:

We sincerely thank the reviewer's comments and suggestions to help improve the manuscript quality. We detail below our responses to the comments and suggestions raised by the reviewer. All the changes in the manuscript have been highlighted in yellow.

COMMENTS: The rebuttal incompletely addresses my concern about the correctness of the field effect mobility expression used in this context. That said, most authors use the same expression for field effect mobility, and the comparison will be useful though not surprising (i.e. a higher mobility away from the interface). I note that Figure R1 assumes that there is only an accumulation layer adjacent the SiO₂ above the back-gate, and not on the exposed or encapsulated surface of the channel. However, the most direct data (ARPES, STS) address the exposed interface, not the buried interface. The manuscript must be more clear in its claims regarding where a SEA exists (at a single interface or two interfaces), and how this affects the interpretation of the transport measurements. This can be addressed by a brief addition to the discussion.

The rebuttal does not establish the influence of the FIB preparation procedure on the MoS₂ properties, but it does show that the difference in fresh and non-fresh flakes is observable despite whatever influence there is.

Response: According to the conclusion made in this manuscript, surface electron accumulation (SEA) exists at both surfaces (including the top and bottom surfaces). Because the applied gate voltage induces space charges in the n^+ -Si/SiO₂ interface, these space charges make much more substantial Coulomb effect on the SiO₂/MoS₂ interface (bottom interface). This is the reason why we assumed that the MoS₂/air interface (top interface), which is far away from the space charges, is not influenced too much by gate voltage, as shown in Figure R1. The statement could be more applicable to the thick flakes as our case with thickness higher than 50 nm. If the thickness is too small, the top interface could exhibit an opposite gate-voltage effect compared to the bottom interface and the overall variation of source-drain currents (from the top and bottom interfaces) induced by the field effect decreases. Accordingly, the dI_{ds}/dV_g or the mobility value is underestimated

for the very thin flakes such as monolayers and few-layers.

The reason to exclude the field effect on the top surface and to adopt the conventional FET model for the mobility calculation in this electron accumulation system has been briefly addressed in the revised manuscript, page 17, lines 1-7.

In addition, to comply with the editorial style and regulations of the journal, we have made some changes in the manuscript formatting, including title, abstract, Figure number, wording, etc. These editorial changes were not highlighted in the revised manuscript.

Figure R1. Schematic band alignment of the MOSFET based on the MoS₂ with surface electron accumulation (a) in equilibrium ($V_g = 0$), (b) at forward gate voltage ($V_g > 0$), and (c) at reverse gate voltage ($V_g < 0$). n^+ -Si/MoS₂ is treated as the metal/semiconductor, E_F^m/E_F^s is the Fermi level of metal/semiconductor, and E_C^s is the conduction band minimum of semiconductor.